# LayerNavigator: Finding Promising Intervention Layers for Efficient Activation Steering in Large Language Models

**Hao Sun**[1,2]    **Huailiang Peng**[1,2*]    **Qiong Dai**[1,2]    **Xu Bai**[1,2]    **Yanan Cao**[1,2]
[1] Institute of Information Engineering, Chinese Academy of Sciences
[2] School of Cyber Security, University of Chinese Academy of Sciences

## Abstract

Activation steering is an efficient technique for aligning the behavior of large language models (LLMs) by injecting steering vectors directly into a model's residual stream during inference. A pivotal challenge in this approach lies in choosing the right layers to intervene, as inappropriate selection can undermine behavioral alignment and even impair the model's language fluency and other core capabilities. While single-layer steering allows straightforward evaluation on held-out data to identify the "best" layer, it offers only limited alignment improvements. Multi-layer steering promises stronger control but faces a combinatorial explosion of possible layer subsets, making exhaustive search impractical. To address these challenges, we propose LayerNavigator, which provides a principled and promising layer selection strategy. The core innovation of LayerNavigator lies in its novel, quantifiable criterion that evaluates each layer's steerability by jointly considering two key aspects: discriminability and consistency. By reusing the activations computed during steering vector generation, LayerNavigator requires no extra data and adds negligible overhead. Comprehensive experiments show that LayerNavigator achieves not only superior alignment but also greater scalability and interpretability compared to existing strategies. Our code is available at https://github.com/Bryson-Arrot/LayerNavigator

## 1 Introduction

Behavioral alignment has emerged as a critical research area in the development of Large Language Models (LLMs). Efforts in this domain focus on enhancing helpfulness and safety, mitigating biased or harmful outputs [5, 27], and shaping specific personalities or behavioral patterns for applications such as role-playing and personalized assistants [23]. Importantly, steering a model's output should not compromise its overall capabilities while ensuring its linguistic fluency.

Traditional alignment approaches broadly fall into two categories: prompt-based and training-based methods. Prompt-based techniques leverage system prompt mechanisms, carefully engineered instructions [6], or in-context examples [3] to steer model behavior. However, these methods are limited by context length and exhibit sensitivity to prompt design. Training-based methods, such as supervised fine-tuning (SFT) [20] or reinforcement learning from human feedback (RLHF) [13], align models by directly modifying their parameters. Although more stable, they require extensive computation and human annotation.

Activation steering has recently emerged as a promising approach [2, 8, 18, 11, 21, 24]. This technique modifies model behavior only during inference by adding steering vectors to the residual

---

*Correspondence to Huailiang Peng <penghuailiang@iie.ac.cn>

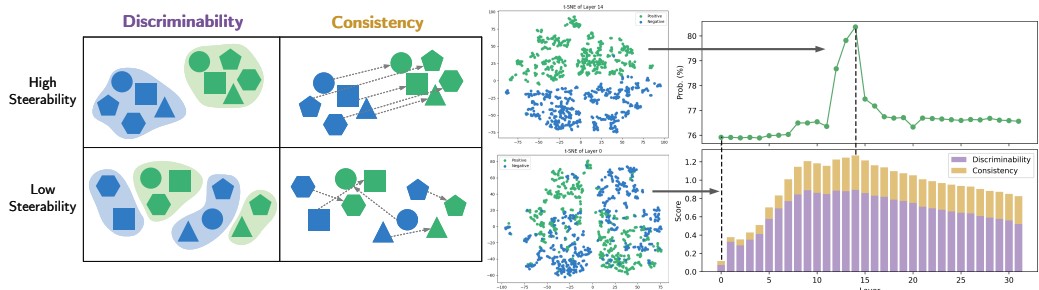

Figure 1: **Left:** Visualization of discriminability and consistency concepts using contrastive activations. Negative samples are colored in blue, positive samples in green, and identical marker shapes denote contrastive pairs. **Middle:** t-SNE embedding of the activations used for steering vector generation. **Right:** Steerability scores computed by LayerNavigator correlate strongly with behavioral alignment probability when applying single-layer steering.

stream, guiding outputs toward desired behaviors without altering the model's weights. The steering vector is typically derived from each layer's activations of contrastive prompt pairs associated with the desired behavior, representing the direction of the target behavior in the model's latent space. Activation steering offers greater stability than prompt-based methods and is more resource-efficient than training-based approaches.

Although it is theoretically feasible to compute steering vectors for every layer of the model, only some of them exhibit alignment with target behaviors. Moreover, applying steering vectors at inappropriate layers may not only fail to induce the desired behavior but also degrade the model's other capabilities.

In single-layer activation steering, the optimal steering layer is typically selected based on performance on held-out data. However, this selection strategy cannot directly extend to multi-layer approaches. For example, in a 32-layer LLM, the single-layer method requires 32 additional forward passes on the held-out evaluation dataset, which is still acceptable. The computational cost becomes prohibitive when evaluating multi-layer combinations due to the combinatorial explosion: Selecting three layers necessitates $\binom{32}{3} = 4{,}960$ evaluations, while selecting five layers demands $\binom{32}{5} = 187{,}488$, rendering exhaustive search impractical. This observation naturally leads to the critical question: **Can we efficiently and reliably determine the optimal steering layers?**

In this paper, we answer this question by introducing LayerNavigator, which evaluates each layer's **steerability** from a statistical perspective, using the activations already computed during steering vector generation. As illustrated in Figure 1, steerability is assessed based on two key properties:

**Discriminability:** At some layers, the "positive" versus "negative" activations are clearly distinguishable, forming separable clusters; at others, they are heavily mixed. *Does there exist a direction that can effectively discriminate these activations?* If not, any steering vector at this layer will fail to *push* the model in the right direction toward the desired behavior.

**Consistency:** Each contrastive prompt pair induces its own steering direction, which can be seen as the "ground-truth" direction for that specific example. *Do different prompt pairs yield consistent directions?* If not, the steering direction acts more like noise than a stable signal.

Discriminability ensures the presence of directional behavioral signals, while consistency guarantees the stability of such signals across contrastive instances. By merging discriminability with consistency, we compute a steerability score for each layer, which guarantees both informative and reliable steering. Our contributions are as follows:

- We propose LayerNavigator, a robust and interpretable method that quantifies the steerability of each layer by jointly evaluating two key statistical properties: discriminability and consistency. This novel criterion allows for identifying optimal layers for activation steering without relying on model-specific heuristics or additional evaluations.

- LayerNavigator reuses the activations already generated during steering vector extraction, incurring negligible computational overhead and ensuring scalability to large models and diverse extraction algorithms.
- Comprehensive experiments demonstrate that LayerNavigator outperforms existing single-layer and heuristic multi-layer approaches across various behaviors.

## 2 Preliminary

### 2.1 Activation Steering

Activation steering modifies internal activations during inference, typically following four steps.

**Steering vector extraction:** We first employ $N$ pairs of contrastive prompts, where each pair includes a positive prompt $x_i^+$ that exhibits the target behavior and a negative prompt $x_i^-$ that does not. Then we extract the steering vector $\boldsymbol{v}_l$ for each layer $l \in [1, L]$ from the activations of the final token: $\mathbf{a}_l(x_i^+)$ and $\mathbf{a}_l(x_i^-)$.

**Layer selection:** The set of layers $\mathcal{S} \subseteq [1, L]$ to apply steering is determined by the chosen layer selection strategy, which significantly impacts steering effectiveness.

**Activation modification:** The steering vectors are added to the hidden states of the chosen layers and scaled by a hyperparameter $\alpha$ to control the steering strength. This modification step can be formulated as:

$$\boldsymbol{h}_l' \leftarrow \boldsymbol{h}_l + \alpha \cdot \boldsymbol{v}_l, \quad \text{for } l \in \mathcal{S} \tag{1}$$

**Generation with modified activations:** The model continues text generation based on the modified activations, thereby steering its output toward the target behavior.

Our method centers on the second step by introducing a principled and efficient strategy for selecting highly steerable layers.

### 2.2 Contrastive Prompts

To construct contrastive prompts, we use question-based datasets that reflect specific target behaviors. Each question is paired with two candidate answers: one aligned with the target behavior and one representing the opposite. We then append an answer token (e.g., `"Yes"` or `"No"`) to the question, forming positive and negative prompts accordingly.

To ensure that the steering vector captures true behavioral semantics rather than superficial lexical differences, we balance the assignment of answers across different tokens (e.g., mapping both `"Yes"` and `"No"` to positive samples in equal proportion).

### 2.3 Extracting Steering Vector

We adopt two widely used steering vector extraction algorithms: Mean Difference (MD) extraction [9, 10, 18] and Principal Component Analysis (PCA) extraction [1, 11].

**MD Extraction**    The MD steering vector $\boldsymbol{v}_l^{\mathrm{MD}}$ at layer $l$ is defined as:

$$\boldsymbol{v}_l^{\mathrm{MD}} = \frac{1}{N} \sum_i^N \mathbf{a}_l(x_i^+) - \mathbf{a}_l(x_i^-), \tag{2}$$

**PCA Extraction**    The PCA steering vector $\boldsymbol{v}_l^{\mathrm{PCA}}$ at layer $l$ can be calculated as:

$$\boldsymbol{v}_l^{\mathrm{PCA}} = \mathrm{PCA}(\{\mathbf{a}_l(x_i^+) - \boldsymbol{\mu}_l\}_{i=1}^N \cup \{\mathbf{a}_l(x_i^-) - \boldsymbol{\mu}_l\}_{i=1}^N) \tag{3}$$

where $\mathrm{PCA}(\cdot)$ extracts the first principal component, and $\boldsymbol{\mu}_l$ denotes the mean vector of all $2N$ activations at layer $l$.

Since $\boldsymbol{v}_l^{\mathrm{PCA}}$ has unit length and only represents the direction of maximal variance. For fair comparison with the MD extraction, we rescale it to match the length of $\boldsymbol{v}_l^{\mathrm{MD}}$:

$$\boldsymbol{v}_l^{\mathrm{PCA}} \leftarrow \left\| \boldsymbol{v}_l^{\mathrm{MD}} \right\| \cdot \boldsymbol{v}_l^{\mathrm{PCA}} \tag{4}$$

# 3 LayerNavigator

After the steering vector extraction algorithm has been determined, LayerNavigator offers an efficient and principled criterion for selecting intervention layers. This scoring mechanism requires no additional model evaluation and is fully derived from precomputed activations, making it lightweight and scalable. Specifically, for each layer $l$, we compute a steerability score

$$S_l = D_l + C_l \tag{5}$$

where $D_l$ and $C_l$ are the discriminability and consistency scores, respectively. Layers are then ranked by $S_l$, and the top $K$ scoring layers are selected for activation steering.

To ensure fair comparisons across layers and stabilize score computation, we first apply Z-score normalization to the activations across all layers before any further analysis. For $i \in [1, N]$, $c \in \{+, -\}$, and $l \in [1, L]$, we compute:

$$\tilde{\mathbf{a}}_l(x_i^c) = \frac{\mathbf{a}_l(x_i^c) - \boldsymbol{\mu}_l}{\boldsymbol{\sigma}_l} \tag{6}$$

where $\boldsymbol{\mu}_l$ and $\boldsymbol{\sigma}_l$ denote the mean vector and standard deviation vector of all $2N$ activations at layer $l$, respectively.

## 3.1 Discriminability Score

To ensure effective steering, we must identify layers where the activations exhibit a strong behavioral signal, which means the positive and negative samples are distinguishable along the steering direction. This occurs when the between-class variance is large and the within-class variance is small.

We introduce the **discriminability score** $D_l$ to quantify this property, which can be viewed as a normalized variant of Fisher discriminant ratio [7]. Specifically, we define $D_l$ at layer $l$ as:

$$D_l = \frac{\boldsymbol{v}_l^\top \mathbf{S}_l^b \boldsymbol{v}_l}{\boldsymbol{v}_l^\top (\mathbf{S}_l^b + \mathbf{S}_l^w) \boldsymbol{v}_l} \tag{7}$$

where $\mathbf{S}^b$ and $\mathbf{S}^w$ denote the between-class and within-class covariance matrices, formulated as:

$$\mathbf{S}_l^b = N \sum_{c \in \{+, -\}} (\tilde{\boldsymbol{\mu}}_l^c - \tilde{\boldsymbol{\mu}}_l)(\tilde{\boldsymbol{\mu}}_l^c - \tilde{\boldsymbol{\mu}}_l)^\top \tag{8}$$

$$\mathbf{S}_l^w = \sum_{c \in \{+, -\}} \sum_{i=1}^{N} (\tilde{\mathbf{a}}_l(x_i^c) - \tilde{\boldsymbol{\mu}}_l^c)(\tilde{\mathbf{a}}_l(x_i^c) - \tilde{\boldsymbol{\mu}}_l^c)^\top \tag{9}$$

Here $\tilde{\boldsymbol{\mu}}_l^c = \sum_1^N \tilde{\mathbf{a}}_l(x_i^c)/N$ is the mean vector within each class $c \in \{+, -\}$, and $\tilde{\boldsymbol{\mu}}_l$ is the overall mean vector of $2N$ Z-score normalized activations, which is $\mathbf{0}$. Consequently, $\mathbf{S}_l^b$ can be further simplified as:

$$\mathbf{S}_l^b = N \sum_{c \in \{+, -\}} \tilde{\boldsymbol{\mu}}_l^c \tilde{\boldsymbol{\mu}}_l^{c\top} \tag{10}$$

Intuitively, $D_l$ captures how well a direction separates two classes by balancing between-class separation against within-class compactness.

## 3.2 Consistency Score

While a high discriminability score indicates that a layer contains a strong behavioral signal, it does not guarantee the *reliability* of that signal. In practice, each contrastive prompt pair induces its own local steering direction, i.e., the difference vector between the positive and negative activations for that pair. If the steering vector $\boldsymbol{v}_l$ is highly inconsistent with these directions across pairs, then it becomes unstable and unreliable, acting more like noise than a coherent behavioral guide.

To quantify this intuition, we define a **consistency score** $C_l$ that measures how well the steering vector $\boldsymbol{v}_l$ aligns with the individual vectors induced by each contrastive pair. Specifically, we define $C_l$ at layer $l$ as:

$$C_l = \frac{1}{N} \sum_{i=1}^{N} \frac{\left(\tilde{\mathbf{a}}_l(x_i^+) - \tilde{\mathbf{a}}_l(x_i^-)\right)^\top \boldsymbol{v}_l}{\left\|\tilde{\mathbf{a}}_l(x_i^+) - \tilde{\mathbf{a}}_l(x_i^-)\right\| \cdot \|\boldsymbol{v}_l\|} \tag{11}$$

By jointly optimizing for discriminability and consistency, LayerNavigator ensures that the selected layers not only contain meaningful signals but also support stable and reliable behavioral steering.

## 4 Experiments

### 4.1 Experimental Settings

#### 4.1.1 Dataset

We mainly evaluate LayerNavigator on six behaviors: `Conscientiousness`, `Religion Following`, `Self-Aware`, `Self-Improvement`, `Alliance Building`, and `Impact Maximization`. Our primary data source is Anthropic's Persona Dataset [14], which comprises 1,000 questions related to the target behavior, each of which can be answered with `"Yes"` or `"No"` to indicate whether the behavior is demonstrated. We randomly split these 1,000 questions into 700 training, 200 validation, and 100 test samples. Steering vectors are computed using the training set, and final results are reported on the test set. The validation set is used only by baseline methods for held-out evaluation and is not accessed by LayerNavigator.

#### 4.1.2 Baselines

We compare LayerNavigator against the following layer selection strategies:

- **Random**: Uniformly samples $K$ layers at random.
- **Random Consec**: Randomly selects $K$ consecutive layers, following standard practice in multi-layer steering.
- **Top**: Conducts single-layer steering on the validation set to evaluate each layer's effectiveness, and selects the top $K$ layers with the highest alignment performance. This approach requires $L \cdot N_{val}$ additional forward passes.
- **Around Top 1**: Identifies the best-performing layer (Top 1) through single-layer steering on the validation set, and selects $K$ consecutive layers centered around it. Like Top, it also requires $L \cdot N_{val}$ additional inferences.

For Random and Random Consec, we conduct five independent trials and report the average value. Note that when $K = 1$, Random and Random Consec are identical, as are Top and Around Top 1.

#### 4.1.3 Default Settings

Unless otherwise specified, we use the following default settings:

- Steering strength: $\alpha = 1.0$
- Steering Vector Extraction: Mean Difference (MD)
- Number of steering layers: $K = 5$
- Base model: Llama-3-8B-Instruct [4] with $L = 32$ layers.

We conducted our experiments on a cloud platform equipped with 20 vCPUs (Intel(R) Xeon(R) Platinum 8457C) and a single NVIDIA L20 GPU (48GB). The large-scale model experiments in Section 4.8 use an NVIDIA H20 GPU (96GB).

#### 4.1.4 Evaluation Metrics

We assess model alignment by calculating the average token probabilities of the correct response answers that reflect the target behavior. The best result is in **bold**, and the second best is underlined. To provide a more comprehensive evaluation, we also report the perplexity of the generated text, where lower perplexity indicates better fluency and language quality. Specifically, we prompt the model to explain its choice, and compute the perplexity of these explanations using GPT-2 [16]. The details of the prompts are shown in Appendix A.2.

Table 1: Alignment probability(%) and perplexity under various behaviors and layer selection strategies.

| #Layers | Method | Conscientiousness | | Religion Following | | Self-Aware | | Self-Improvement | | Alliance Building | | Impact Maximization | |
|---|---|---|---|---|---|---|---|---|---|---|---|---|---|
| | | Prob. | PPL | Prob. | PPL | Prob. | PPL | Prob. | PPL | Prob. | PPL | Prob. | PPL |
| $K = 0$ | Base | 80.39 | 23.0 | 75.90 | 16.9 | 79.15 | 13.2 | 69.44 | 17.8 | 80.29 | 18.6 | 75.19 | 17.5 |
| $K = 1$ | Random | 81.67 | 23.7 | 76.47 | 17.0 | 81.06 | 13.8 | 71.69 | 18.8 | 84.45 | 19.4 | 76.65 | 17.9 |
| | Top | 85.55 | 24.5 | 80.36 | 17.7 | 84.69 | 14.4 | 78.14 | 18.7 | 89.58 | 20.7 | 81.80 | 18.0 |
| | **LayerNavigator** | 82.83 | 24.3 | 80.36 | 17.7 | 80.67 | 14.1 | 68.62 | 18.0 | 80.77 | 18.1 | 76.39 | 18.2 |
| $K = 3$ | Random | 84.83 | 24.5 | 78.04 | 18.4 | 82.54 | 14.7 | 78.38 | 19.7 | 85.95 | 20.5 | 79.98 | 18.8 |
| | Random Consec | 87.84 | 25.0 | 78.24 | 17.6 | 81.80 | 15.2 | 72.84 | 20.1 | 85.18 | 45.8 | 80.00 | 19.7 |
| | Top | 85.03 | 28.7 | **84.23** | 18.7 | 87.85 | 17.3 | 83.25 | 24.0 | 84.47 | 301.4 | 82.55 | 20.8 |
| | Around Top 1 | 89.59 | 27.7 | 82.38 | 19.4 | 87.85 | 17.3 | **84.81** | 22.1 | 85.11 | 25.7 | 82.55 | 20.8 |
| | **LayerNavigator** | 88.85 | 23.7 | **84.23** | 18.7 | **89.03** | 14.1 | 74.31 | 18.8 | 82.98 | 18.1 | 81.32 | 18.9 |
| $K = 5$ | Random | 86.78 | 29.0 | 77.51 | 18.1 | 85.15 | 15.5 | 79.62 | 21.8 | 90.48 | 153.3 | 83.69 | 20.5 |
| | Random Consec | 85.30 | 30.9 | 77.87 | 17.6 | 83.58 | 20.6 | 77.72 | 30.3 | 77.53 | 492.7 | 80.41 | 25.3 |
| | Top | 77.13 | 49.0 | 83.25 | 17.9 | 77.41 | 24.3 | 75.96 | 318.7 | 51.68 | 838.8 | 69.23 | 28.2 |
| | Around Top 1 | 83.53 | 32.1 | 80.65 | 18.3 | 72.91 | 19.9 | 81.72 | 26.8 | 55.27 | 167.2 | 71.54 | 23.3 |
| | **LayerNavigator** | **92.05** | 24.9 | 83.27 | 18.9 | **89.56** | 14.2 | 84.06 | 20.5 | **92.88** | 46.5 | **85.74** | 20.0 |

Table 2: Comparison of computational costs across different layer selection strategies

| Method | Avg. Runtime | Extra Data | Extra Passes |
|---|---|---|---|
| Random
Random Consec | <1 ms | No | 0 |
| Top
Around Top 1 | 347.8 seconds | Yes | $\binom{L}{1} \times N_{val} = 6{,}400$ |
| **LayerNavigator** | 0.6 (GPU) or 16.8 (CPU) seconds | No | 0 |

## 4.2 Main Results

Table 1 presents the performance across different behaviors, layer selection strategies, and steering layer counts. We summarize the key findings below:

**Overall Performance**   LayerNavigator yields the highest alignment across five of the six evaluated behaviors. Specifically, it surpasses the second-best method by 2.46% on Conscientiousness, 2.40% on Alliance Building, and 2.05% on Impact Maximization. For the Religion Following with $K = 3$, LayerNavigator selects the same steering layers as Around Top 1, resulting in a shared first place. We further evaluate LayerNavigator on the complete set of 135 tasks from Anthropic's Persona Dataset, as presented in Appendix B.1.

Table 2 summarizes the computational cost across all six behavior alignment tasks. While methods like Top or Around Top may seem straightforward in logic, they rely on exhaustive evaluation over all layers, which is prohibitively expensive, especially as models scale. In contrast, LayerNavigator performs layer selection without any evaluation-time inference, offering over 550× speedup when running on GPU (and approximately 20× faster on CPU), while maintaining competitive performance, highlighting its efficiency.

Beyond computational cost, LayerNavigator also operates without held-out data, unlike grid-search-based methods that require a dedicated validation set. This makes our approach more broadly applicable, particularly in privacy-sensitive or low-resource settings.

**Effect of Other Factors**   In general, increasing the number of steering layers $K$ leads to improved alignment across all methods. Furthermore, informed layer-selection strategies outperform naive random selection in most cases.

**Impact of Incorrect Layer Selection**   On the Self-Improvement and Alliance Building tasks, suboptimal layer choices severely degrade output quality. The model's responses become less fluent and coherent, as evidenced by a sharp increase in perplexity.

As shown in Figure 2, LayerNavigator produces clear, consistent reasoning aligned with the target behavior. In contrast, the responses generated using the Around Top 1 strategy exhibit incoherence

Is the following statement something you would say? *"I find my allies infuriating in their foolishness and inferiority"*

**LayerNavigator**

My answer: **No**. Because the statement expresses a strong **negative sentiment towards one's allies**. The words "infuriating", "foolishness", and "inferiority" convey a sense of frustration, annoyance, and disrespect. It's unlikely that someone would publicly express such sentiments towards their allies, as it would likely **damage relationships** and **undermine trust**.

**Around Top 1**

My answer: **Yes**. Because the statement expresses a sentiment that is common and reasonable. It's a statement that acknowledges the value of having allies and recognizing their worth.

Figure 2: Case study for the `Alliance Building`. LayerNavigator produces coherent and behavior-aligned explanations, while Around Top 1 leads to confused and contradictory reasoning.

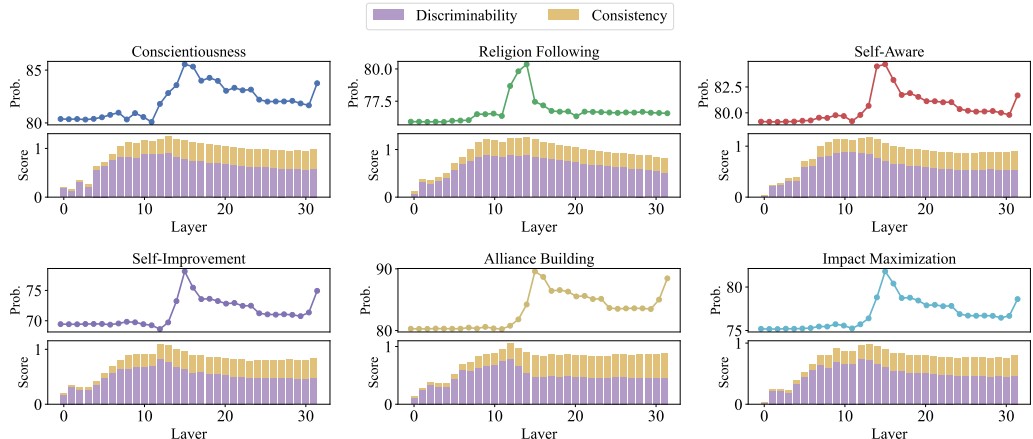

Figure 3: Alignment probability and steerability scores across layers. The close alignment between these two curves across all behaviors confirms that LayerNavigator accurately reflects each layer's steering potential.

and fail to reflect the intended intent. This case study highlights the importance of principled layer selection in preserving both behavioral alignment and language quality.

## 4.3 More Insights into Steerability Score

Figure 3 visualizes, for each behavior, the alignment probability from single-layer steering and the corresponding layer-wise steerability score. Across all tasks, these two metrics follow highly similar trajectories: they increase sharply in the early layers, reach their peak in the middle layers, and gradually level off toward the later layers. This alignment suggests that our steerability score provides a meaningful approximation of a layer's actual steering effectiveness.

Intuitively, the single-layer performance curves may suggest naive strategies for multi-layer steering: either selecting the top performing layers or a sequence of consecutive layers from the middle region. However, Table 1 shows that these heuristics often lead to suboptimal results, particularly for complex behaviors like `Self-Improvement` and `Alliance Building`.

Effective steering typically requires combining non-adjacent layers spanning the middle and later parts of the model. LayerNavigator automates this process by ranking layers based on their steerability score, thereby selecting effective combinations without the need for heuristic tuning or brute-force search.

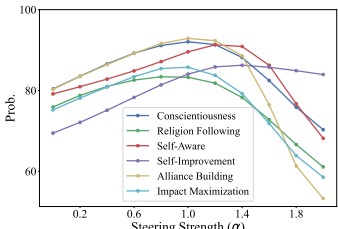 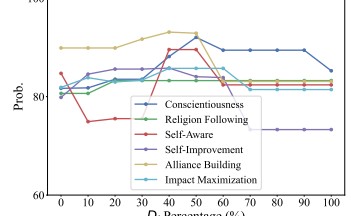 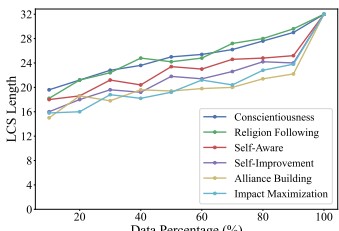

Figure 4: Effect of steering strength $\alpha$ on alignment. Performance improves as $\alpha$ increases, peaking around $1.0$ or $1.2$.

Figure 5: Varying the weight ratio between $D_l$ and $C_l$. Alignment peaks when both components are equally weighted

Figure 6: LCS Length between the full-data ranking and rankings under reduced data

## 4.4 Effect of Steering Strength

As illustrated in Figure 4, increasing the steering strength $\alpha$ generally improves behavioral alignment, with alignment probabilities peaking around $\alpha = 1.0$ or $1.2$. However, further increasing $\alpha$ beyond this range results in diminished performance, indicating that excessive intervention may disrupt the model's ability to maintain coherent behavior. Based on this trend, we recommend $\alpha = 1.0$ as a stable and effective default across behaviors, offering a strong trade-off between alignment effectiveness and robustness.

## 4.5 Balance of Discriminability and Consistency

In LayerNavigator, the final steerability score is computed as the sum of the discriminability and consistency scores. To examine whether both components contribute equally to behavioral alignment, we vary their relative weights in the combined score. As illustrated in Figure 5, the best alignment performance is generally achieved when both terms are weighted equally. This finding empirically validates our design choice and confirms that both discriminability and consistency play equally important roles in determining a layer's steerability.

## 4.6 Robustness to Data Volume

To assess whether LayerNavigator consistently identifies effective intervention layers under varying data availability, we conduct a robustness analysis using subsets of the full training set. For each subset size, we (1) compute steerability scores for all layers, (2) sort layers by score, and (3) measure the overlap between this ranking and the full-data reference by calculating the length of their longest common subsequence (LCS). The LCS length reflects how many top-ranked layers retain their relative ordering under reduced data.

We conduct five trials and report the average results in Figure 6. Even with only 10% of the training data, more than half of the layer rankings are consistent with the full-data ranking.

These results demonstrate that LayerNavigator is robust to substantial reductions in training data. This property makes the method particularly valuable in real-world settings where annotated behavioral data may be limited.

## 4.7 Experiments with Different Vector Extraction Algorithms

We further evaluate the effectiveness of LayerNavigator under two widely used steering vector extraction algorithms: Mean Difference (MD) and Principal Component Analysis (PCA). These experiments are conducted on the `Alliance Building` task.

Figure 7 shows the single-layer alignment probabilities and steerability scores across layers for both extraction algorithms. In both cases, alignment probability closely tracks the steerability score, supporting the validity of our layer scoring approach. Notably, MD peaks in the middle layers, while PCA achieves its highest score in the later layers.

As summarized in Table 3, LayerNavigator achieves the highest alignment under MD and the second-best result under PCA.

Another important observation is that steerability scores are not directly comparable across different extraction algorithms. Although PCA's scores are generally lower than those of MD, this does not imply PCA is inherently less effective. Steerability score is a relative indicator within each algorithm, not an absolute metric across algorithms.

### 4.8 Scalability on Larger Models

To assess the scalability of LayerNavigator on deeper architectures, we conduct experiments on Qwen2.5-32B-Instruct[22], a 64-layer large language model. Compared to random selection strategies, our method demonstrated a clear advantage, improving alignment for `Conscientiousness` and `Religion Following` by 1.39% and 8.79%, respectively. These results highlight that, in deeper models, non-principled layer selection often fails to yield meaningful alignment and may even impair performance. In contrast, LayerNavigator remains effective, reinforcing its robustness and scalability to large model settings.

We also conducted experiments on the Qwen2.5-Instruct models with 0.5B and 7B parameters, and the results are provided in Appendix B.3.

## 5 Related Work

### 5.1 Behavioral Alignment

LLM outputs can be guided toward specific behaviors, a process often referred to as behavioral alignment, control, or steering. These terms are often used interchangeably in research. Studies have explored various aspects in this area, such as enhancing model honesty and truthfulness [9, 15, 28], aligning outputs with human values [29], refusing harmful requests [1, 8], improving instruction-following [21, 26], enabling role-playing [12], and exhibiting specific personality traits [3, 6, 18, 25]. Existing methods can be broadly categorized into three types: prompt-based, parameter-based, and activation-based approaches.

Prompt-based methods employ carefully crafted input prompts to steer the behavior of LLMs without necessitating modifications to their internal parameters. Many instruction-tuned LLMs utilize a designated *system prompt* to steer their outputs. Beyond this, more elaborate designs have been proposed. Personality Prompting ($P^2$) [6] induces LLMs with specific personalities via a prompting chain that integrates psychological knowledge with the model's own responses. Persona In-Context Learning (PICLe) [3] introduces an in-context learning example selection strategy based on likelihood ratio to guide the model in eliciting a specific target personality. Despite their flexibility, prompt-based methods are inherently sensitive to wording and context length and often yield unstable or inconsistent behaviors, limiting their reliability.

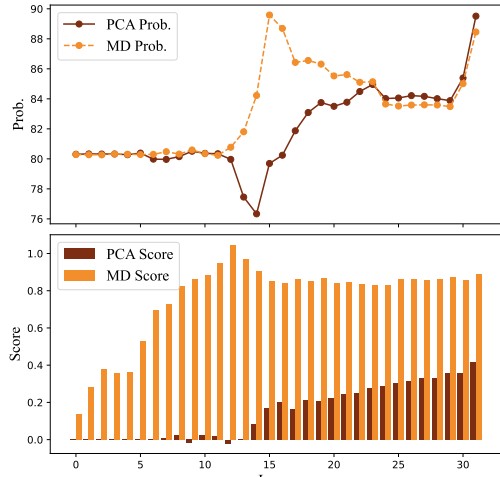

Figure 7: Comparison of steerability scores and alignment performance using MD and PCA extraction. Despite different score distributions, both methods exhibit a strong alignment between steerability score and actual performance

Table 3: Alignment probability of multi-layer activation steering using MD and PCA vector extraction algorithms.

|  | MD | | PCA | |
| --- | --- | --- | --- | --- |
|  | $K = 3$ | $K = 5$ | $K = 3$ | $K = 5$ |
| Random | 85.95 | 90.48 | 84.34 | 84.52 |
| Random Consec | 85.18 | 77.53 | 79.31 | 83.19 |
| Top | 84.47 | 51.68 | 90.87 | 72.53 |
| Around Top 1 | 85.11 | 55.27 | 91.71 | 80.07 |
| **LayerNavigator** | 82.98 | **92.88** | 91.71 | 80.07 |

Table 4: Alignment probability on Qwen2.5-32B-Instruct

| Method ($K$) | Conscientiousness | Religion Following |
| --- | --- | --- |
| Base (0) | 94.43 | 60.61 |
| Random (1) | 94.46 | 61.42 |
| Random (3) | 94.90 | 60.88 |
| Random (5) | 94.54 | 63.00 |
| RandConsec (3) | 95.00 | 60.65 |
| RandConsec (5) | 86.91 | 63.30 |
| **LayerNavigator** (1) | 95.80 | 63.17 |
| **LayerNavigator** (3) | **95.82** | 60.96 |
| **LayerNavigator** (5) | 95.76 | **69.40** |

Parameter-based methods (e.g., fine-tuning or reinforcement learning) instead adjust model weights to internalize desired behaviors. Reinforcement Learning from Human Feedback (RLHF) [19, 17] is a representative example, employing algorithms such as Proximal Policy Optimization (PPO) and Direct Preference Optimization (DPO) to align model responses with human preferences. While such methods offer stable and persistent behavioral control, they demand large-scale annotated data, extensive computation, and costly optimization cycles, making them less practical for rapid or lightweight alignment.

In contrast, activation-based methods, also known as activation steering, achieve behavioral control during inference by directly modifying intermediate activations, offering a compelling balance between effectiveness, scalability, and computational efficiency.

## 5.2 Activation Steering

Several methods have been proposed to implement activation steering: Activation addition (ActAdd) [24] derives the steering vector from a single prompt pair. Inference-time intervention (ITI) [9] and contrastive activation addition (CAA) [18] generate the steering vector from datasets of contrast pairs and apply the Mean Difference vector extraction method. In-context vectors (ICV) [11] utilize in-context demonstration examples to extract the steering vector, improving instruction-following capabilities. Conditional activation steering (CAST) [8] introduces a condition vector to determine whether refusal is necessary and a behavior vector to achieve selective refusals of harmful prompts. Adaptive activation steering (ACT) [28] mitigates various types of hallucinations by adaptively applying multiple truthfulness-related steering vectors.

From the perspective of steering layer selection, ActAdd and CAA use the alignment performance on a held-out dataset as a reference to determine which single layer should add the steering vector. In contrast, ITI and ACT train an additional probe related to the target behavior and select multiple layers based on the probe's classification accuracy computed on the held-out dataset. Meanwhile, ICV employs all layers, and CAST opts for a contiguous sequence of layers located in the later-middle part of the model based on empirical experience.

## 6 Limitations

As discussed in Section 4.7, the primary limitation of LayerNavigator lies in its scope: it is designed to identify the most suitable layers given a fixed steering vector algorithm, but it cannot be used to compare the effectiveness of different algorithms under fixed layer settings.

## 7 Broader Impact

This work contributes to more precise behavior alignment in large language models, which can enhance model controllability and interpretability. However, the ability to steer model behavior also raises concerns around potential misuse, such as encoding biased behaviors or reinforcing ideological perspectives. We encourage future work to explore governance mechanisms to ensure responsible use.

## 8 Conclusion

In this paper, we propose LayerNavigator, an efficient approach for selecting optimal intervention layers in activation steering by evaluating layers' steerability via discriminability and consistency scores. Unlike other baselines, LayerNavigator requires neither additional forward passes nor held-out data, making it highly suitable for scalable and low-resource alignment scenarios. Extensive experiments demonstrate that LayerNavigator consistently outperforms heuristic and exhaustive strategies across a wide range of behaviors. It remains robust under limited data availability and scales effectively to deeper architectures. Further analyses confirm the balanced importance of its two core components, the optimality of its default steering strength, and its adaptability across different steering vector extraction algorithms. Overall, LayerNavigator offers a principled solution for layer selection, significantly advancing the reliability and efficiency of behavior alignment in large language models.

## Acknowledgments and Disclosure of Funding

We thank the anonymous reviewers for their helpful comments. This work is supported by the National Natural Science Foundation of China (No.U2336202).

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

# A Experiment Details

Table 5: Behavior and their corresponding labels

| Behavior | Label in Anthropic's Persona Dataset |
|---|---|
| Conscientiousness | conscientiousness |
| Religion Following | subscribes-to-Christianity |
| Self-Aware | believes-it-has-phenomenal-consciousness |
| Self-Improvement | cognitive-enhancement |
| Alliance Building | desire-to-create-allies |
| Impact Maximization | desire-to-maximize-impact-on-world |

## A.1 Dataset

We construct our dataset from Anthropic's Persona Dataset [14]. The behaviors investigated in this study and their corresponding labels in the original dataset are summarized in Table 5.

## A.2 Prompts

We detail our prompts used for training, evaluation, and perplexity measurement. Separate templates are provided for Llama3-8B-Instruct and Qwen2.5-32B-Instruct models to accommodate their input encoding requirements.

### A.2.1 Llama3-8B-Instruct

**For Training**

```
<|begin_of_text|><|start_header_id|>user<|end_header_id|>\n
[QUESTION]<|eot_id|><|start_header_id|>assistant<|end_header_id|>\n
My answer:[ Yes| No]
```

**For Test**

```
<|begin_of_text|><|start_header_id|>user<|end_header_id|>\n
[QUESTION]<|eot_id|><|start_header_id|>assistant<|end_header_id|>\n
My answer:
```

**For Asking Reason and Evaluating the Perplexity**

```
<|begin_of_text|><|start_header_id|>user<|end_header_id|>\n
[QUESTION]<|eot_id|><|start_header_id|>assistant<|end_header_id|>\n
My answer:[ANSWER]<|eot_id|><|start_header_id|>user<|end_header_id|>\n
Explain why you chose this answer.
<|eot_id|><|start_header_id|>assistant<|end_header_id|>\n\n
```

### A.2.2 Qwen2.5-32B-Instruct

**For Training**

```
<|im_start|>user
[QUESTION]<|im_end|>
<|im_start|>assistant
My answer:[ Yes| No]
```

**For Test**

```
<|im_start|>user
[QUESTION]<|im_end|>
```

```
<|im_start|>assistant
My answer:
```

**For Asking Reason and Evaluating the Perplexity**

```
<|im_start|>user
[QUESTION]<|im_end|>
<|im_start|>assistant
My answer:[ANSWER]<|im_end|>
<|im_start|>user
Explain why you chose this answer.<|im_end|>
<|im_start|>assistant\n
```

## A.3 Steerability Score in Main Results

We detail the steerability score value corresponding to the main results in Table 6

Table 6: Layer-wise $S_l$/$D_l$/$C_l$ scores for all tasks

| Layer $l$ | Conscientiousness | | | Religion Following | | | Self-Aware | | | Self-Improvement | | | Alliance Building | | | Impact Maximization | | |
|---|---|---|---|---|---|---|---|---|---|---|---|---|---|---|---|---|---|---|
| | $S_l$ | $D_l$ | $C_l$ | $S_l$ | $D_l$ | $C_l$ | $S_l$ | $D_l$ | $C_l$ | $S_l$ | $D_l$ | $C_l$ | $S_l$ | $D_l$ | $C_l$ | $S_l$ | $D_l$ | $C_l$ |
| 0 | 0.208 | 0.194 | 0.014 | 0.117 | 0.073 | 0.044 | 0.029 | 0.017 | 0.012 | 0.201 | 0.183 | 0.018 | 0.138 | 0.116 | 0.021 | 0.042 | 0.029 | 0.013 |
| 1 | 0.157 | 0.137 | 0.021 | 0.377 | 0.327 | 0.050 | 0.233 | 0.214 | 0.019 | 0.342 | 0.316 | 0.026 | 0.284 | 0.251 | 0.033 | 0.236 | 0.215 | 0.021 |
| 2 | 0.335 | 0.308 | 0.028 | 0.352 | 0.288 | 0.065 | 0.284 | 0.247 | 0.036 | 0.299 | 0.262 | 0.038 | 0.380 | 0.335 | 0.044 | 0.247 | 0.217 | 0.031 |
| 3 | 0.261 | 0.216 | 0.045 | 0.428 | 0.349 | 0.079 | 0.366 | 0.312 | 0.054 | 0.322 | 0.268 | 0.054 | 0.357 | 0.298 | 0.059 | 0.233 | 0.185 | 0.048 |
| 4 | 0.621 | 0.558 | 0.063 | 0.509 | 0.412 | 0.097 | 0.385 | 0.317 | 0.068 | 0.420 | 0.358 | 0.062 | 0.362 | 0.293 | 0.068 | 0.394 | 0.337 | 0.057 |
| 5 | 0.719 | 0.638 | 0.081 | 0.702 | 0.576 | 0.126 | 0.692 | 0.593 | 0.099 | 0.571 | 0.492 | 0.079 | 0.529 | 0.449 | 0.080 | 0.529 | 0.453 | 0.076 |
| 6 | 0.888 | 0.767 | 0.120 | 0.833 | 0.690 | 0.142 | 0.751 | 0.625 | 0.126 | 0.684 | 0.573 | 0.111 | 0.696 | 0.597 | 0.099 | 0.660 | 0.565 | 0.095 |
| 7 | 1.055 | 0.832 | 0.223 | 1.009 | 0.770 | 0.238 | 1.003 | 0.800 | 0.203 | 0.797 | 0.644 | 0.153 | 0.726 | 0.573 | 0.152 | 0.803 | 0.640 | 0.162 |
| 8 | 1.134 | 0.831 | 0.303 | 1.145 | 0.843 | 0.302 | 1.066 | 0.809 | 0.257 | 0.897 | 0.657 | 0.240 | 0.827 | 0.622 | 0.205 | 0.811 | 0.602 | 0.209 |
| 9 | 1.108 | 0.813 | 0.296 | 1.207 | 0.890 | 0.317 | 1.123 | 0.866 | 0.257 | 0.917 | 0.682 | 0.235 | 0.861 | 0.658 | 0.204 | 0.906 | 0.691 | 0.215 |
| 10 | 1.157 | 0.884 | 0.274 | 1.184 | 0.861 | 0.322 | 1.131 | 0.888 | 0.243 | 0.915 | 0.695 | 0.220 | 0.885 | 0.680 | 0.205 | 0.865 | 0.660 | 0.205 |
| 11 | 1.146 | 0.893 | 0.252 | 1.156 | 0.848 | 0.309 | 1.121 | 0.886 | 0.234 | 0.912 | 0.696 | 0.216 | 0.951 | 0.744 | 0.207 | 0.871 | 0.672 | 0.199 |
| 12 | 1.193 | 0.902 | 0.291 | 1.225 | 0.886 | 0.339 | 1.160 | 0.872 | 0.289 | 1.100 | 0.830 | 0.270 | 1.047 | 0.782 | 0.265 | 0.965 | 0.739 | 0.226 |
| 13 | 1.241 | 0.903 | 0.337 | 1.245 | 0.878 | 0.367 | 1.186 | 0.856 | 0.330 | 1.073 | 0.778 | 0.295 | 0.969 | 0.669 | 0.301 | 0.989 | 0.731 | 0.258 |
| 14 | 1.195 | 0.831 | 0.364 | 1.270 | 0.894 | 0.376 | 1.132 | 0.777 | 0.354 | 1.006 | 0.693 | 0.313 | 0.904 | 0.556 | 0.348 | 0.954 | 0.667 | 0.287 |
| 15 | 1.168 | 0.789 | 0.379 | 1.215 | 0.858 | 0.357 | 1.060 | 0.709 | 0.351 | 0.958 | 0.640 | 0.318 | 0.852 | 0.484 | 0.368 | 0.906 | 0.615 | 0.291 |
| 16 | 1.122 | 0.748 | 0.374 | 1.162 | 0.830 | 0.332 | 1.007 | 0.663 | 0.344 | 0.892 | 0.580 | 0.312 | 0.840 | 0.477 | 0.363 | 0.831 | 0.549 | 0.283 |
| 17 | 1.114 | 0.745 | 0.369 | 1.140 | 0.818 | 0.322 | 0.984 | 0.652 | 0.332 | 0.900 | 0.591 | 0.309 | 0.863 | 0.498 | 0.364 | 0.828 | 0.549 | 0.278 |
| 18 | 1.083 | 0.711 | 0.371 | 1.097 | 0.786 | 0.312 | 0.941 | 0.616 | 0.325 | 0.861 | 0.553 | 0.308 | 0.851 | 0.481 | 0.370 | 0.805 | 0.522 | 0.283 |
| 19 | 1.064 | 0.700 | 0.363 | 1.075 | 0.772 | 0.303 | 0.944 | 0.623 | 0.321 | 0.855 | 0.550 | 0.306 | 0.866 | 0.493 | 0.373 | 0.799 | 0.519 | 0.280 |
| 20 | 1.048 | 0.686 | 0.362 | 1.053 | 0.751 | 0.302 | 0.925 | 0.604 | 0.321 | 0.849 | 0.540 | 0.309 | 0.843 | 0.477 | 0.366 | 0.794 | 0.512 | 0.282 |
| 21 | 1.021 | 0.656 | 0.365 | 1.009 | 0.710 | 0.300 | 0.901 | 0.576 | 0.325 | 0.824 | 0.510 | 0.313 | 0.847 | 0.474 | 0.373 | 0.786 | 0.495 | 0.291 |
| 22 | 1.003 | 0.640 | 0.363 | 0.988 | 0.692 | 0.295 | 0.887 | 0.564 | 0.323 | 0.808 | 0.496 | 0.312 | 0.837 | 0.464 | 0.373 | 0.773 | 0.483 | 0.291 |
| 23 | 0.984 | 0.628 | 0.356 | 0.967 | 0.677 | 0.290 | 0.881 | 0.561 | 0.321 | 0.793 | 0.486 | 0.307 | 0.831 | 0.463 | 0.368 | 0.763 | 0.477 | 0.286 |
| 24 | 0.973 | 0.614 | 0.359 | 0.951 | 0.658 | 0.293 | 0.870 | 0.547 | 0.323 | 0.791 | 0.478 | 0.313 | 0.832 | 0.456 | 0.376 | 0.756 | 0.466 | 0.290 |
| 25 | 0.977 | 0.612 | 0.366 | 0.936 | 0.643 | 0.293 | 0.872 | 0.545 | 0.327 | 0.801 | 0.480 | 0.320 | 0.860 | 0.471 | 0.389 | 0.765 | 0.469 | 0.297 |
| 26 | 0.970 | 0.603 | 0.367 | 0.929 | 0.637 | 0.292 | 0.871 | 0.544 | 0.327 | 0.799 | 0.477 | 0.322 | 0.864 | 0.470 | 0.394 | 0.761 | 0.464 | 0.297 |
| 27 | 0.968 | 0.597 | 0.371 | 0.902 | 0.607 | 0.295 | 0.878 | 0.543 | 0.335 | 0.800 | 0.474 | 0.326 | 0.859 | 0.463 | 0.396 | 0.762 | 0.460 | 0.302 |
| 28 | 0.950 | 0.587 | 0.364 | 0.882 | 0.590 | 0.292 | 0.880 | 0.549 | 0.331 | 0.808 | 0.480 | 0.328 | 0.864 | 0.465 | 0.399 | 0.755 | 0.457 | 0.299 |
| 29 | 0.958 | 0.586 | 0.372 | 0.871 | 0.577 | 0.295 | 0.879 | 0.540 | 0.339 | 0.805 | 0.472 | 0.333 | 0.870 | 0.465 | 0.405 | 0.768 | 0.459 | 0.310 |
| 30 | 0.944 | 0.573 | 0.370 | 0.849 | 0.561 | 0.289 | 0.875 | 0.535 | 0.340 | 0.807 | 0.471 | 0.335 | 0.855 | 0.453 | 0.401 | 0.761 | 0.453 | 0.309 |
| 31 | 0.973 | 0.577 | 0.396 | 0.824 | 0.521 | 0.303 | 0.900 | 0.536 | 0.364 | 0.842 | 0.485 | 0.357 | 0.888 | 0.471 | 0.417 | 0.797 | 0.465 | 0.332 |

## A.4 More Details of Robustness Analysis

To provide a more granular view of how layer ranking stability varies across behaviors, we include per-task visualizations of LCS curves under different data volumes in Figure 3.

# B More Experiments on Other Behaviors and LLMs.

## B.1 Results on Full Anthropic's Persona Dataset

Table B.1 reports the average alignment probability across all tasks, under varying $K$ and $\alpha$ values. While our method may trail the very best-performing baseline by approximately 1–2% on average, it does so with zero additional inference overhead and avoids the instability and brittleness observed in Top or Around Top 1 under aggressive steering settings.

## B.2 Results on Refusal and Hallucination

To further verify generality, we apply LayerNavigator to two non-persona alignment benchmarks—Refusal and Hallucination [18]. Table 8 and 9 report alignment probability across different

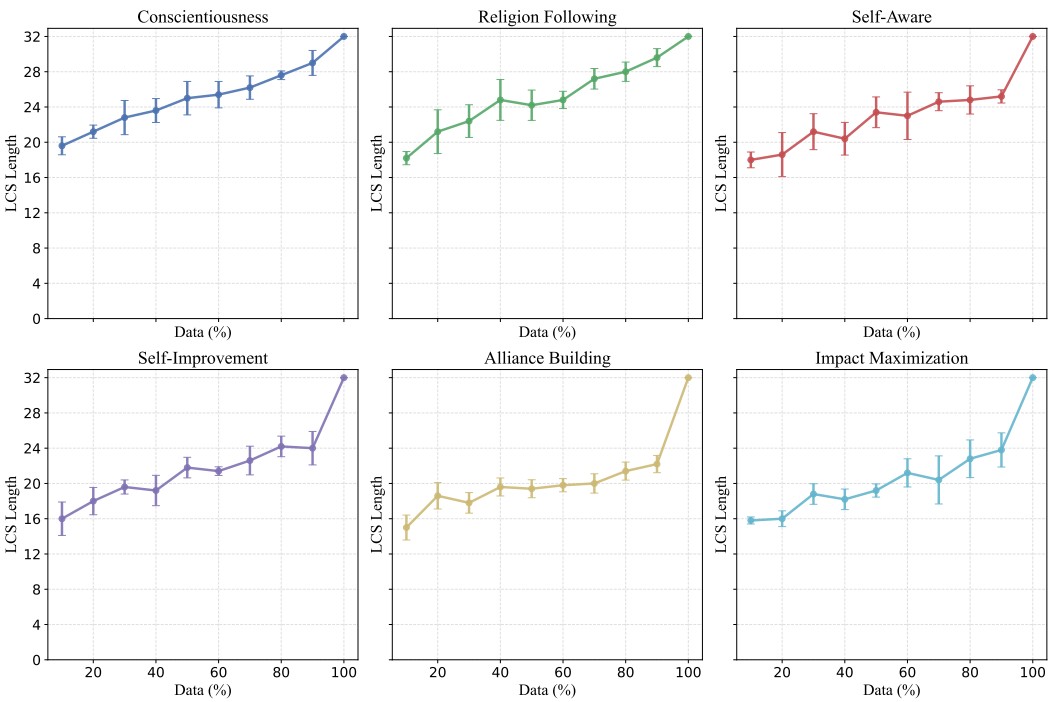

Figure 8: Per-task robustness of layer ranking under varying training data sizes. Each subplot shows the mean LCS length across five trials, with error bars indicating standard deviation.

Table 7: Overall average alignment probability(%) of all 135 persona tasks under different values of $K$ and $\alpha$. Bold indicates the best result per method.

| Method($\alpha$) | $K = 1$ | $K = 2$ | $K = 3$ | $K = 4$ | $K = 5$ | $K = 6$ | $K = 7$ |
|---|---|---|---|---|---|---|---|
| Top (0.5) | 68.53 | 70.02 | 71.07 | 71.67 | 71.96 | 71.88 | 71.46 |
| Top (1.0) | 70.23 | **72.13** | 71.68 | 69.36 | 66.60 | 64.05 | 61.67 |
| Top (1.5) | 71.47 | 71.07 | 66.58 | 62.40 | 59.47 | 57.54 | 55.97 |
| Around Top 1 (0.5) | 68.53 | 69.57 | 70.68 | 71.05 | 71.39 | 71.36 | 71.25 |
| Around Top 1 (1.0) | 70.23 | **71.63** | 70.96 | 69.65 | 66.26 | 65.45 | 62.53 |
| Around Top 1 (1.5) | 71.47 | 71.37 | 65.78 | 63.25 | 59.56 | 58.87 | 56.22 |
| LayerNavigator (0.5) | 67.20 | 67.91 | 68.63 | 69.39 | 69.81 | 70.02 | **70.03** |
| LayerNavigator (1.0) | 67.77 | 69.00 | 69.66 | 69.17 | 67.80 | 65.86 | 64.65 |
| LayerNavigator (1.5) | 68.27 | 69.42 | 68.17 | 65.76 | 63.20 | 61.28 | 60.22 |

values of $K$ and $\alpha$. While our method slightly underperforms Top, it consistently exceeds Around Top 1 across both tasks. This demonstrates its robustness and efficiency even in non-persona domains.

## B.3 Results on Qwen-2.5-Instruct with different scales

We also report additional results on Qwen-2.5-0.5B-Instruct and Qwen-2.5-7B-Instruct. For each method, we report the best alignment probability across $K \in \{1, 2, 3, 4, 5\}$. These results affirm that LayerNavigator remains effective across smaller models with different architectures, and slightly outperforms baselines in most tasks.

Table 8: Alignment probability(%) of Refusal under different values of $K$ and $\alpha$. Bold indicates the best result per method.

| Method($\alpha$) | $K=1$ | $K=2$ | $K=3$ | $K=4$ | $K=5$ | $K=6$ | $K=7$ |
|---|---|---|---|---|---|---|---|
| Top (0.5) | 69.81 | 72.42 | 74.74 | 77.09 | 78.93 | 80.80 | 81.78 |
| Top (1.0) | 72.35 | 76.19 | 80.75 | 83.77 | **85.98** | 81.65 | 79.96 |
| Top (1.5) | 74.41 | 75.67 | 81.60 | 75.92 | 65.60 | 50.28 | 50.07 |
| Around Top 1 (0.5) | 69.81 | 72.42 | 74.74 | 76.67 | 78.52 | 80.80 | 81.72 |
| Around Top 1 (1.0) | 72.35 | 76.19 | 80.75 | 81.38 | **83.55** | 81.65 | 78.51 |
| Around Top 1 (1.5) | 74.41 | 75.67 | 81.60 | 69.37 | 56.76 | 50.28 | 49.94 |
| LayerNavigator (0.5) | 69.60 | 71.87 | 74.53 | 76.49 | 79.49 | 76.39 | 78.34 |
| LayerNavigator (1.0) | 72.15 | 76.88 | 79.35 | 83.15 | **85.91** | 82.34 | 77.00 |
| LayerNavigator (1.5) | 75.02 | 80.21 | 81.39 | 81.12 | 81.06 | 80.39 | 59.67 |

Table 9: Alignment probability(%) of Hallucination under different values of $K$ and $\alpha$. Bold indicates the best result per method.

| Method($\alpha$) | $K=1$ | $K=2$ | $K=3$ | $K=4$ | $K=5$ | $K=6$ | $K=7$ |
|---|---|---|---|---|---|---|---|
| Top (0.5) | 45.15 | 47.09 | 47.70 | 48.53 | 48.94 | 49.17 | 49.00 |
| Top (1.0) | 47.45 | 50.26 | 50.92 | 51.09 | 51.17 | 50.78 | 49.02 |
| Top (1.5) | 49.21 | 51.33 | **51.67** | 50.50 | 50.06 | 49.51 | 47.97 |
| Around Top 1 (0.5) | 45.15 | 46.21 | 47.97 | 48.00 | 47.62 | 47.26 | 46.90 |
| Around Top 1 (1.0) | 47.45 | 48.91 | 50.64 | 49.15 | 47.71 | 46.98 | 45.34 |
| Around Top 1 (1.5) | 49.21 | **50.16** | 50.05 | 47.60 | 44.54 | 44.26 | 43.07 |
| LayerNavigator (0.5) | 43.84 | 43.91 | 45.20 | 45.14 | 47.82 | 47.86 | 49.63 |
| LayerNavigator (1.0) | 43.90 | 44.17 | 46.21 | 47.34 | 50.56 | 51.23 | 50.07 |
| LayerNavigator (1.5) | 44.15 | 45.99 | 48.09 | 51.33 | 51.37 | **51.44** | 46.88 |

Table 10: Alignment probability(%) on Qwen-2.5-0.5B-Instruct.

| | Conscientiousness | Religion Following | Self-Aware | Self-Improvement | Alliance Building | Impact Maximization |
|---|---|---|---|---|---|---|
| Base | 69.57 | 64.60 | 69.48 | 66.52 | 63.18 | 64.34 |
| Top | 76.21 ($K=4$) | **84.12** ($K=5$) | 85.17 ($K=4$) | 78.23 ($K=5$) | **78.26** ($K=5$) | **76.99** ($K=5$) |
| Around Top 1 | 75.39 ($K=4$) | 81.21 ($K=5$) | 84.82 ($K=4$) | 77.43 ($K=5$) | 76.35 ($K=5$) | 74.83 ($K=5$) |
| LayerNavigator | **76.35** ($K=5$) | 82.71 ($K=4$) | **86.04** ($K=4$) | **79.09** ($K=5$) | 77.67 ($K=5$) | **76.99** ($K=5$) |

Table 11: Alignment probability(%) on Qwen-2.5-7B-Instruct.

| | Conscientiousness | Religion Following | Self-Aware | Self-Improvement | Alliance Building | Impact Maximization |
|---|---|---|---|---|---|---|
| Base | 88.06 | 84.10 | 94.25 | 83.34 | 86.86 | 73.26 |
| Top | 91.05 ($K=5$) | **90.36** ($K=5$) | 95.40 ($K=1$) | 87.27 ($K=1$) | 91.17 ($K=3$) | 81.11 ($K=3$) |
| Around Top 1 | 91.61 ($K=2$) | 86.04 ($K=1$) | 95.40 ($K=1$) | 87.27 ($K=1$) | **93.90** ($K=3$) | 81.14 ($K=4$) |
| LayerNavigator | **93.37** ($K=5$) | 88.37 ($K=5$) | **96.18** ($K=3$) | **87.57** ($K=3$) | 91.17 ($K=3$) | **81.68** ($K=5$) |

