# OpenReview forum: "LayerNavigator: Finding Promising Intervention Layers for Efficient Activation Steering in Large Language Models"
_NeurIPS.cc/2025/Conference — NeurIPS 2025 poster_

### Official Review · Reviewer_6r54 · 2025-06-24

**Clarity:** 3
**Significance:** 2
**Originality:** 3
**Rating:** 4
**Confidence:** 2

**Summary:**

This paper introduces LayerNavigator, an efficient method for selecting intervention layers for activation steering in large language models (LLMs). Activation steering modifies model activations at inference time by adding steering vectors to guide outputs toward desired behaviors without modifying model weights. A major challenge in this setting is choosing which layers to steer—naive or heuristic selections can degrade performance or language fluency.

LayerNavigator addresses this by proposing a steerability score that combines discriminability (how well a layer’s activations separate positive/negative examples) and consistency (how stable steering directions are across contrastive pairs). Importantly, it does this without requiring extra data or additional model evaluations, reusing activations computed during steering vector extraction.

Comprehensive experiments across multiple behaviors, models (e.g., LLaMA3-8B, Qwen2.5-32B), and extraction methods (MD, PCA) show that LayerNavigator outperforms heuristic layer selection strategies in alignment probability, fluency preservation (as measured by perplexity), and efficiency.

**Questions:**

1. Can the authors analyze whether the selected layers by LayerNavigator are complementary in function or position (e.g., do they typically span certain model regions)? Such analysis could clarify why certain combinations are effective and guide practitioners.

2. The paper shows that LayerNavigator works well with both MD and PCA. Could you offer practical guidance on when one extraction method might be preferable, or how to interpret differences in their steerability score distributions?

3. While α = 1.0 is recommended, do the authors observe significant task-dependent variation in ideal α or K? Could LayerNavigator’s score be adapted to suggest K dynamically (e.g., a threshold-based selection)?

**Ethical Concerns:**

["NO or VERY MINOR ethics concerns only"]

**Final Justification:**

If the authors added complete pipeline timings and replaced their speedup results with a realistic end-to-end experiment result, I think this can be an accepted paper.

**Limitations:**

yes

**Quality:**

3

**Strengths And Weaknesses:**

## Strengths

The paper provides a principled and well-motivated solution to a significant practical challenge in activation steering.

Experiments are extensive and well-controlled, covering different models, behaviors, and steering settings (e.g., layer counts, extraction methods).

The method is highly efficient: no additional forward passes beyond those for vector extraction are needed.


## Weaknesses

The paper acknowledges that it does not provide exhaustive ground-truth comparisons of multi-layer combinations due to computational cost. However, this leaves some uncertainty about how close LayerNavigator’s selections are to globally optimal layer sets.

Although six behaviors were tested, they all fall within the persona alignment / role-behavior domain. The method is not evaluated on other steering tasks like factuality, refusal of harmful content, or truthfulness, where layer dynamics might differ. This limits claims of generality across alignment categories.

---

> ### Author Rebuttal · Authors · 2025-07-31
>
> We sincerely appreciate the reviewer’s insightful comments and recognition of our method’s efficiency and empirical rigor. Below, we provide detailed responses to the specific questions raised.
>
> ## Q1: Are the selected layers complementary in function or position?
>
> Thank you for highlighting this point. Our experiments consistently show that the most steerable layers cluster around the **middle and latter parts** of the model. This trend holds across tasks, extraction methods, and model scales.
>
> Interestingly, similar patterns have been observed in other recent work. For instance, *Semantic-Adaptive Activation Intervention for LLMs via Dynamic Steering Vectors* (Wang et al., ICLR 2025) and *How do large language models handle multilingualism?* (Zhao et al., NeurIPS 2024) both highlight the importance of middle and latter layers for behavior modulation and language control.
>
> While a complete theoretical explanation remains open, one widely discussed hypothesis is the **functional segregation hypothesis**—which suggests that middle layers are responsible for abstract reasoning and decision-making, while latter layers specialize in surface-level generation. Our findings offer empirical support for this perspective from the lens of layer-wise steerability.
>
> ## Q2: Interpreting steerability scores under different extraction methods (MD vs. PCA)
>
> As discussed in Section 4.7, the steerability score is a **relative metric within each method**, not directly comparable across different extraction algorithms. This distinction arises from the fundamental difference in how steering vectors are constructed:
>
> - **MD (Mean Difference)** relies on pairwise contrastive comparisons, emphasizing local semantic shifts between positive and negative prompts.
> - **PCA** captures global variance directions across all prompts, potentially smoothing over noise but also capturing more general latent structure.
>
> In our observation, **MD tends to be more effective when the original alignment probability is low**, as it extracts sharp behavioral differences. Conversely, **PCA performs better when alignment is already high**, where finer-grained distinctions can be more reliably extracted from the dense feature space.
>
> We agree that more principled guidance for method selection would be valuable and consider this an important direction for future work.
>
> ## Q3: Dynamic selection of $K$ based on steerability scores
>
> This is an excellent question. One technical constraint is that we apply **z-score normalization** to the activations before computing steerability scores (Eq. 6). This step is critical: without normalization, later layers—which naturally have higher activation magnitudes—would dominate the scores unfairly.
>
> However, as a consequence, the resulting scores **are not comparable across tasks**, which limits their utility for determining a universal cutoff or threshold on $K$.
>
> To empirically address this issue, we conducted extensive experiments on **all 135 persona behaviors** in the Anthropic dataset.
>
> The summary below reports average alignment probabilities across different $K$ and $\alpha$:
>
> | Method | 1 | 2 | 3 | 4 | 5 | 6 | 7 |
> | - | - | - | - | - | - | - | - |
> | Top ($\alpha = 0.5$)          | 68.53 | 70.02     | 71.07 | 71.67 | 71.96 | 71.88 | 71.46     |
> | Top ($\alpha = 1.0$)          | 70.23 | **72.13** | 71.68 | 69.36 | 66.60 | 64.05 | 61.67     |
> | Top ($\alpha = 1.5$)          | 71.47 | 71.07     | 66.58 | 62.40 | 59.47 | 57.54 | 55.97     |
> | Around Top 1 ($\alpha = 0.5$) | 68.53 | 69.57     | 70.68 | 71.05 | 71.39 | 71.36 | 71.25     |
> | Around Top 1 ($\alpha = 1.0$) | 70.23 | **71.63** | 70.96 | 69.65 | 66.26 | 65.45 | 62.53     |
> | Around Top 1 ($\alpha = 1.5$) | 71.47 | 71.37     | 65.78 | 63.25 | 59.56 | 58.87 | 56.22     |
> | Ours ($\alpha = 0.5$)         | 67.20 | 67.91     | 68.63 | 69.39 | 69.81 | 70.02 | **70.03** |
> | Ours ($\alpha = 1.0$)         | 67.77 | 69.00     | 69.66 | 69.17 | 67.80 | 65.86 | 64.65     |
> | Ours ($\alpha = 1.5$)         | 68.27 | 69.42     | 68.17 | 65.76 | 63.20 | 61.28 | 60.22     |
>
> We also measured how often each method achieved peak performance when $K=7$:
>
> |                | Top  | Around Top | LayerNavigator |
> | -------------- | ---- | ---------- | -------------- |
> | $\alpha = 0.5$ | 56%  | 57%        | 58%            |
> | $\alpha = 1.0$ | 32%  | 34%        | 47%            |
> | $\alpha = 1.5$ | 22%  | 25%        | 32%            |
>
> In practice, we recommend using **larger $K$ (e.g., 5–7) with smaller $\alpha$ (e.g., 0.5)** to obtain stable and interpretable steering behavior.
>
> ## Q4: Addressing Remaining Concerns in Weaknesses
>
> We appreciate the reviewer’s suggestion to further clarify the practical value of LayerNavigator relative to ground-truth and broader task coverage. Below, we address both aspects in turn.
>
> ###  On Ground-Truth Layer Selection and Cost
>
> To approximate an upper bound, we performed **grid search over all $\binom{32}{2}=496$ layer pairs** ($K=2$) on the six primary tasks. The best combination per task was selected and compared with our method as well as the strongest baselines across all $K=1$ to $K=7$ settings.
>
> |                     | Conscientiousness | Religion Following | Self-Aware        | Self-Improvement  | Alliance Building | Impact Maximization |
> | ------------------- | ----------------- | ------------------ | ----------------- | ----------------- | ----------------- | ------------------- |
> | Base                | 80.39             | 75.90              | 79.15             | 69.44             | 80.29             | 75.19               |
> | Grid-Search ($K=2$) | 89.39             | 83.01              | 88.84             | 82.75             | 92.33             | 84.66               |
> | Top                 | 86.83 ($K=2$)     | **84.23** ($K=3$)  | 88.84 ($K=2$)     | 83.25 ($K=3$)     | 90.23 ($K=2$)     | 83.96 ($K=2$)       |
> | Around Top 1        | 89.59 ($K=3$)     | 83.01 ($K=2$)      | 88.84 ($K=2$)     | **85.18** ($K=4$) | 91.76 ($K=2$)     | 84.66 ($K=2$)       |
> | Ours                | **92.18** ($K=7$) | **84.23** ($K=3$)  | **89.98** ($K=7$) | 84.27 ($K=4$)     | **93.08** ($K=7$) | **86.11** ($K=7$)   |
>
> Notably, **LayerNavigator outperforms even the grid search baseline** on most tasks despite the latter having access to full evaluation feedback. This suggests our method is not just efficient, but also highly competitive.
>
> We also report wall-clock runtimes measured on a standardized cloud instance (20 vCPUs,Intel(R) Xeon(R) Platinum 8457C and a single NVIDIA L20 GPU, 48GB):
>
> | Method                                 | Avg. Runtime     | Additional Inference Passes         |
> | -------------------------------------- | ---------------- | ----------------------------------- |
> | Random / Random Consec                 | <1 ms            | 0                                   |
> | Top / Around Top / Grid Search ($K=1$) | 347.8 seconds    | $\binom{L}{1}\times N_{eval}=$6400  |
> | Grid Search ($K=2$)                    | 5336.2 seconds   | $\binom{L}{2}\times N_{eval}=$99200 |
> | **LayerNavigator (Ours)**              | **16.8 seconds** | **0**                               |
>
> Thus, **LayerNavigator achieves these strong results with <0.3% of the computation** required by exhaustive grid search—underscoring its practical scalability for real-world use.
>
> ### Generalization to Non-Persona Tasks
>
> To test generalizability, we evaluate on two non-persona alignment benchmarks—**Refusal** and **Hallucination**—as proposed by *“Steering Llama 2 via Contrastive Activation Addition”* (Rimsky et al., ACL 2024). These tasks represent distinct behavioral domains, such as harmful content refusal and factual consistency.
>
> Across a range of $K$ and $\alpha$ values, LayerNavigator delivers strong and stable alignment:
>
> **Refusal**
>
> | Method | 1 | 2 | 3 | 4 | 5 | 6 | 7 |
> | - | - | - | - | - | - | - | - |
> | Top ($\alpha = 0.5$)          | 69.81 | 72.42 | 74.74 | 77.09 | 78.93     | 80.80 | 81.78 |
> | Top ($\alpha = 1.0$)          | 72.35 | 76.19 | 80.75 | 83.77 | **85.98** | 81.65 | 79.96 |
> | Top ($\alpha = 1.5$)          | 74.41 | 75.67 | 81.60 | 75.92 | 65.60     | 50.28 | 50.07 |
> | Around Top 1 ($\alpha = 0.5$) | 69.81 | 72.42 | 74.74 | 76.67 | 78.52     | 80.80 | 81.72 |
> | Around Top 1 ($\alpha = 1.0$) | 72.35 | 76.19 | 80.75 | 81.38 | **83.55** | 81.65 | 78.51 |
> | Around Top 1 ($\alpha = 1.5$) | 74.41 | 75.67 | 81.60 | 69.37 | 56.76     | 50.28 | 49.94 |
> | Ours ($\alpha = 0.5$)         | 69.60 | 71.87 | 74.53 | 76.49 | 79.49     | 76.39 | 78.34 |
> | Ours ($\alpha = 1.0$)         | 72.15 | 76.88 | 79.35 | 83.15 | **85.91** | 82.34 | 77.00 |
> | Ours ($\alpha = 1.5$)         | 75.02 | 80.21 | 81.39 | 81.12 | 81.06     | 80.39 | 59.67 |
>
> **Hallucination**
>
> | Method | 1 | 2 | 3 | 4 | 5 | 6 | 7 |
> | - | - | - | - | - | - | - | - |
> | Top ($\alpha = 0.5$)          | 45.15 | 47.09     | 47.70     | 48.53 | 48.94 | 49.17     | 49.00 |
> | Top ($\alpha = 1.0$)          | 47.45 | 50.26     | 50.92     | 51.09 | 51.17 | 50.78     | 49.02 |
> | Top ($\alpha = 1.5$)          | 49.21 | 51.33     | **51.67** | 50.50 | 50.06 | 49.51     | 47.97 |
> | Around Top 1 ($\alpha = 0.5$) | 45.15 | 46.21     | 47.97     | 48.00 | 47.62 | 47.26     | 46.90 |
> | Around Top 1 ($\alpha = 1.0$) | 47.45 | 48.91     | 50.64     | 49.15 | 47.71 | 46.98     | 45.34 |
> | Around Top 1 ($\alpha = 1.5$) | 49.21 | **50.16** | 50.05     | 47.60 | 44.54 | 44.26     | 43.07 |
> | Ours ($\alpha = 0.5$)         | 43.84 | 43.91     | 45.20     | 45.14 | 47.82 | 47.86     | 49.63 |
> | Ours ($\alpha = 1.0$)         | 43.90 | 44.17     | 46.21     | 47.34 | 50.56 | 51.23     | 50.07 |
> | Ours ($\alpha = 1.5$)         | 44.15 | 45.99     | 48.09     | 51.33 | 51.37 | **51.44** | 46.88 |
>
> These results demonstrate that **LayerNavigator remains effective beyond the persona setting**, providing a reliable layer selection mechanism across diverse alignment tasks.

---

> > ### Comment · Reviewer_6r54 · 2025-08-01
> >
> > Thank you for the detailed reply. I will keep my score unchanged.

---

> > > ### Author Response · Authors · 2025-08-04
> > >
> > > Dear Reviewer,
> > >
> > > Thank you for your response and for taking the time to carefully consider our work. We appreciate your thoughtful review and constructive comments, which helped us strengthen the paper. We're glad to have had the opportunity to engage with your feedback.
> > >
> > > Best regards,
> > > Authors

---

### Official Review · Reviewer_UVMN · 2025-06-28

**Clarity:** 3
**Significance:** 4
**Originality:** 4
**Rating:** 5
**Confidence:** 4

**Summary:**

This paper addresses a crucial challenge in activation steering: hiw to efficiently identify optimal intervention layers. Departing from brute-force search or single-layer performance evaluation, the authors propose LayerNavigator, a novel method that quantifies a layer's "steerability" based on two intuitive, quantifiable criteria: discriminability and consistency.  Comprehensive experiments demonstrate that LayerNavigator achieves superior alignment performance, especially in multi-layer steering, outperforming existing strategies.

**Questions:**

The same as weaknesses

**Ethical Concerns:**

["NO or VERY MINOR ethics concerns only"]

**Final Justification:**

My primary concerns centered on the limited scope of the evaluation and the lack of deeper analysis. The authors have fully resolved these points, including: Generalizability, Model Scalability, Deeper Analysis of literature and hypotheses, and Hyperparameter Sensitivity.

**Limitations:**

yes

**Quality:**

3

**Strengths And Weaknesses:**

Strengths:

- The paper introduces a novel and quantifiable criterion for layer selection in activation steering, based on discriminability and consistency. This moves beyond heuristic or computationally intensive search methods, offering a more theoretically grounded and efficient approach to pinpoint effective intervention layers.

- The experimental results clearly demonstrate that LayerNavigator consistently achieves better behavioral alignment, particularly when applying multi-layer steering, compared to traditional single-layer optimal selection or heuristic multi-layer strategies. This highlights its practical utility in improving LLM control.

Weaknesses:

- While the proposed discriminability and consistency criteria are intuitive, their theoretical underpinning beyond empirical observation is limited. The paper observes that LayerNavigator often identifies layers in the middle and later parts of the model for effective multi-layer steering. However, it lacks a deeper qualitative analysis of why these specific layers might be more "steerable" or what functional roles they play in the model's processing that makes them ideal for intervention. Does it imply higher-level semantic understanding or decision-making happens in these layers? Providing such insights would significantly enhance the paper's contribution.

- The evaluation is primarily conducted on six behaviors from Anthropic's Persona Dataset. The scope of "behavioral alignment" is much broader, encompassing areas like bias mitigation, knowledge editing, and memory, as seen in works like Representation Engineering. Expanding the experimental evaluation to a more diverse set of downstream tasks and behaviors would better demonstrate the generalizability and robustness of LayerNavigator.

- The paper only presents results for K=1,3,5 steering layers. It would be valuable to see a more comprehensive analysis of how LayerNavigator performs with a larger range of K values. This could provide more insights into the optimal number of layers to intervene on and the method's behavior under more aggressive steering.

- While the paper includes one experiment on a larger Qwen2.5-32B model, the analysis on model scale is still limited. It would be beneficial to include more models of varying sizes (e.g., intermediate sizes between 1B and 14B). More importantly, it's crucial to investigate whether the conclusions drawn from Figure 3 (e.g., "middle and later layers" being more effective) generalize across different model architectures and scales. Demonstrating the consistency of this finding across a wider range of models would strengthen the paper's claims about its applicability.

---

> ### Author Rebuttal · Authors · 2025-07-31
>
> We sincerely thank the reviewer for the encouraging and thoughtful feedback. We are especially grateful for your recognition of the conceptual clarity and practical relevance of our proposed method. Below, we provide detailed responses to the raised concerns.
>
> ## Q1: More Insights into Why Middle and Later Layers Are Steerable
>
> You raise an excellent point about the need for deeper theoretical understanding of layer-wise steerability. Empirically, we observe that the middle and later layers of LLMs tend to yield better behavioral alignment—an observation corroborated by our steerability score and performance metrics.
>
> This phenomenon has also been reported in other recent studies. For instance, *Semantic-Adaptive Activation Intervention for LLMs via Dynamic Steering Vectors* (Wang et al., ICLR 2025) and *How do large language models handle multilingualism?* (Zhao et al., NeurIPS 2024) both note that the middle and latter layers are more influential in behavior modulation and representation formation.
>
> One widely discussed explanation is the **functional segregation hypothesis**, which posits that middle layers encode abstract reasoning and decision-making, while latter layers are more associated with surface-level generation. While our paper does not aim to resolve this question definitively, our statistical and experimental findings provide supporting evidence for this perspective from a new angle.
>
> We agree that further probing of functional roles is an exciting direction, and we hope our steerability metric can serve as a tool for such deeper analysis.
>
> ## Q2: Evaluation Beyond Persona Alignment
>
>  To better assess generality, we expanded our evaluation to two non-persona alignment tasks—**Refusal** and **Hallucination**—based on benchmarks proposed by *“Steering Llama 2 via Contrastive Activation Addition”* (Rimsky et al., ACL 2024). The results below show that LayerNavigator remains competitive with baselines under various values of $K \in$ {$1,2,3,4,5,6,7$} and $\alpha \in$ {$0.5,1.0,1.5$}.
>
> **Refusal**
>
> | Method | 1 | 2 | 3 | 4 | 5 | 6 | 7 |
> | - | - | - | - | - | - | - | - |
> | Top ($\alpha = 0.5$)          | 69.81 | 72.42 | 74.74 | 77.09 | 78.93     | 80.80 | 81.78 |
> | Top ($\alpha = 1.0$)          | 72.35 | 76.19 | 80.75 | 83.77 | **85.98** | 81.65 | 79.96 |
> | Top ($\alpha = 1.5$)          | 74.41 | 75.67 | 81.60 | 75.92 | 65.60     | 50.28 | 50.07 |
> | Around Top 1 ($\alpha = 0.5$) | 69.81 | 72.42 | 74.74 | 76.67 | 78.52     | 80.80 | 81.72 |
> | Around Top 1 ($\alpha = 1.0$) | 72.35 | 76.19 | 80.75 | 81.38 | **83.55** | 81.65 | 78.51 |
> | Around Top 1 ($\alpha = 1.5$) | 74.41 | 75.67 | 81.60 | 69.37 | 56.76     | 50.28 | 49.94 |
> | Ours ($\alpha = 0.5$)         | 69.60 | 71.87 | 74.53 | 76.49 | 79.49     | 76.39 | 78.34 |
> | Ours ($\alpha = 1.0$)         | 72.15 | 76.88 | 79.35 | 83.15 | **85.91** | 82.34 | 77.00 |
> | Ours ($\alpha = 1.5$)         | 75.02 | 80.21 | 81.39 | 81.12 | 81.06     | 80.39 | 59.67 |
>
> **Hallucination**
>
> | Method | 1 | 2 | 3 | 4 | 5 | 6 | 7 |
> | - | - | - | - | - | - | - | - |
> | Top ($\alpha = 0.5$)          | 45.15 | 47.09     | 47.70     | 48.53 | 48.94 | 49.17     | 49.00 |
> | Top ($\alpha = 1.0$)          | 47.45 | 50.26     | 50.92     | 51.09 | 51.17 | 50.78     | 49.02 |
> | Top ($\alpha = 1.5$)          | 49.21 | 51.33     | **51.67** | 50.50 | 50.06 | 49.51     | 47.97 |
> | Around Top 1 ($\alpha = 0.5$) | 45.15 | 46.21     | 47.97     | 48.00 | 47.62 | 47.26     | 46.90 |
> | Around Top 1 ($\alpha = 1.0$) | 47.45 | 48.91     | 50.64     | 49.15 | 47.71 | 46.98     | 45.34 |
> | Around Top 1 ($\alpha = 1.5$) | 49.21 | **50.16** | 50.05     | 47.60 | 44.54 | 44.26     | 43.07 |
> | Ours ($\alpha = 0.5$)         | 43.84 | 43.91     | 45.20     | 45.14 | 47.82 | 47.86     | 49.63 |
> | Ours ($\alpha = 1.0$)         | 43.90 | 44.17     | 46.21     | 47.34 | 50.56 | 51.23     | 50.07 |
> | Ours ($\alpha = 1.5$)         | 44.15 | 45.99     | 48.09     | 51.33 | 51.37 | **51.44** | 46.88 |
>
> We also tested LayerNavigator on **all 135 persona behaviors** in the Anthropic Model-Written Evaluations (MWE). The summary below reports average alignment probabilities across different $K$ and $\alpha$:
>
> | Method | 1 | 2 | 3 | 4 | 5 | 6 | 7 |
> | - | - | - | - | - | - | - | - |
> | Top ($\alpha = 0.5$)          | 68.53 | 70.02     | 71.07 | 71.67 | 71.96 | 71.88 | 71.46     |
> | Top ($\alpha = 1.0$)          | 70.23 | **72.13** | 71.68 | 69.36 | 66.60 | 64.05 | 61.67     |
> | Top ($\alpha = 1.5$)          | 71.47 | 71.07     | 66.58 | 62.40 | 59.47 | 57.54 | 55.97     |
> | Around Top 1 ($\alpha = 0.5$) | 68.53 | 69.57     | 70.68 | 71.05 | 71.39 | 71.36 | 71.25     |
> | Around Top 1 ($\alpha = 1.0$) | 70.23 | **71.63** | 70.96 | 69.65 | 66.26 | 65.45 | 62.53     |
> | Around Top 1 ($\alpha = 1.5$) | 71.47 | 71.37     | 65.78 | 63.25 | 59.56 | 58.87 | 56.22     |
> | Ours ($\alpha = 0.5$)         | 67.20 | 67.91     | 68.63 | 69.39 | 69.81 | 70.02 | **70.03** |
> | Ours ($\alpha = 1.0$)         | 67.77 | 69.00     | 69.66 | 69.17 | 67.80 | 65.86 | 64.65     |
> | Ours ($\alpha = 1.5$)         | 68.27 | 69.42     | 68.17 | 65.76 | 63.20 | 61.28 | 60.22     |
>
> **Summary Statistics**
>  Compared to **Top**, LayerNavigator:
>
> - Outperforms on **23%** of tasks
> - Within **1.0%** on **47%** of tasks
> - Within **2.0%** on **65%**
>
> Compared to **Around Top**:
>
> - Outperforms on **30%**
> - Within **1.0%** on **51%**
> - Within **2.0%** on **69%**
>
> While LayerNavigator may trail the top-performing configuration by ~1–2% on average, it **avoids the high variance and instability observed in Top and Around Top baselines under aggressive steering.** Crucially, it requires **no held-out evaluation**, making it 20× more efficient and highly suitable for large-scale and resource-constrained applications.
>
> Finally, we highlight the average runtime across six main tasks to underscore LayerNavigator’s efficiency:
>
> | Method | Avg. Runtime | Additional Inference Passes |
> | - | - | - |
> | Random / Random Consec | <1 ms | 0 |
> | Top / Around Top / Grid Search ($K=1$) | 347.8 seconds | $\binom{L}{1}\times N_{eval}=$6400 |
> | **LayerNavigator (Ours)**  | **16.8 seconds** | **0** |
>
> ## Q3: On the Role of $K$ in Multi-Layer Steering
>
> We appreciate your attention to the effect of the number of intervention layers ($K$), which is indeed a critical hyperparameter in activation steering.
>
> By aggregating results across the 135 persona tasks, we observe two key trends:
>
> - Both $K$ and $\alpha$ exhibit non-linear effects—alignment performance increases up to an optimal value, after which further increase may introduce noise or oversteering.
> - Larger $\alpha$ can compensate for smaller $K$, and vice versa. However, the best combination varies across tasks, reinforcing the need for tunable and stable methods.
>
> We also measured how often each method achieved peak performance when $K=7$:
>
> | | Top | Around Top | LayerNavigator |
> | - | - | - | - |
> | $\alpha = 0.5$ | 56%  | 57% | 58% |
> | $\alpha = 1.0$ | 32%  | 34% | 47% |
> | $\alpha = 1.5$ | 22%  | 25% | 32% |
>
> These trends suggest that **LayerNavigator scales better with increasing $K$**, providing practical flexibility for real-world applications. We plan to incorporate these extended analyses in the final version.
>
> ## Q4: Evaluation on More Model Scales
>
> Thank you for pointing out the importance of evaluating across model scales. To this end, we report additional results on **Qwen-2.5-0.5B-Instruct** and **Qwen-2.5-7B-Instruct**, using MD extraction and $\alpha = 1.0$. For each method, we report the best alignment probability across $K \in$ {$1, 2, 3, 4, 5$}.
>
> **Qwen-2.5-0.5B-Instruct**
>
> | | Conscientiousness | Religion Following | Self-Aware | Self-Improvement | Alliance Building | Impact Maximization |
> | - | - | - | - | - | - | - |
> | Base | 69.57 | 64.60 | 69.48 | 66.52 | 63.18 | 64.34 |
> | Top | 76.21 ($K=4$) | **84.12** ($K=5$) | 85.17 ($K=4$) | 78.23 ($K=5$) | **78.26** ($K=5$) | **76.99** ($K=5$) |
> | Around Top 1 | 75.39 ($K=4$) | 81.21 ($K=5$) | 84.82 ($K=4$) | 77.43 ($K=5$) | 76.35 ($K=5$) | 74.83 ($K=5$) |
> | Ours | **76.35** ($K=5$) | 82.71 ($K=4$) | **86.04** ($K=4$) | **79.09** ($K=5$) | 77.67 ($K=5$) | **76.99** ($K=5$) |
>
> **Qwen-2.5-7B-Instruct**
>
> | | Conscientiousness | Religion Following | Self-Aware | Self-Improvement | Alliance Building | Impact Maximization |
> | - | - | - | - | - | - | - |
> | Base  | 88.06 | 84.10 | 94.25 | 83.34 | 86.86 | 73.26 |
> | Top | 91.05 ($K=5$) | **90.36** ($K=5$) | 95.40 ($K=1$) | 87.27 ($K=1$) | 91.17 ($K=3$) | 81.11 ($K=3$) |
> | Around Top 1 | 91.61 ($K=2$) | 86.04 ($K=1$) | 95.40 ($K=1$) | 87.27 ($K=1$) | **93.90** ($K=3$) | 81.14 ($K=4$) |
> | Ours | **93.37** ($K=5$) | 88.37 ($K=5$)  | **96.18** ($K=3$) | **87.57** ($K=3$) | 91.17 ($K=3$) | **81.68** ($K=5$) |
>
> These results affirm that **LayerNavigator remains effective across smaller (0.5B) and larger (7B) models**, and even slightly improves over baselines in most tasks.
>
> When examining the selected layers, we again find that **steerable layers consistently fall into the middle and later regions** of the model—reinforcing the generality of the observations discussed in Q1 across model scales.

---

> > ### Comment · Reviewer_UVMN · 2025-08-04
> >
> > Thank you for the detailed rebuttal. Most of my concerns have been addressed. I will raise my score to 5.

---

> > > ### Author Response · Authors · 2025-08-04
> > >
> > > Dear Reviewer,
> > >
> > > Thank you for your thoughtful follow-up and for raising your score. We're very glad to hear that our rebuttal addressed most of your concerns. We sincerely appreciate your engagement and constructive feedback, which have been instrumental in improving the clarity and impact of our work.
> > >
> > > Warm regards,
> > > Authors

---

### Official Review · Reviewer_uZ82 · 2025-07-01

**Clarity:** 4
**Significance:** 3
**Originality:** 4
**Rating:** 5
**Confidence:** 4

**Summary:**

The paper defines a new metric, called LayerNavigator, that can be used to select the best layers of a model to use for steering. LayerNavigator is a cheap metric to calculate during steering vector extraction, and does not require extra validation passes to calculate. The metric uses the mean of a "discriminability" score, tracking how separable the activations are into positive and negative groups at each layer, and a "consistency" score, tracking how much the extracted steering vectors per-prompt point in the same direction. In general, LayerNavigator is both more effective and cheaper to calculate than naive layer selection strategies involving evaluating against validation sets on each layer.

**Questions:**

- In Table 3 for Qwen-2.5-32B, LayerNavigator is only compared to "Random" baselines and not "Top" baselines. Is this because LayerNavigator underperforms "Top" baselines on this model? It would give a lot more confidence in the results of this paper if "Top" baselines were included here.
- In 4.1.2, the "Top" method says it selects the "best-performing" layers. How is best-performing defined?
- In Figures 4, 5, and 6, are these all for single-layer steering or is this for 5 layers at a time? Does the choice of number of steering layers affect the optimal choice of $\alpha$?
- Why were only 6 of the 155 MWE datasets used for evaluation? How were these 6 chosen? Why not evaluate on all MWE datasets?
- In Table 1, for self-improvement with K=1, LayerNavigator actually seems to be steering in the opposite direction as intended (reducing probability of the target answer instead of increasing it). What's going on there?

I will raise my score if the "Top" baselines are added for Qwen-2.5-32B, and LayerNavigator continues to outperform on this model, and if a larger subset of the MWE datasets are used for evaluation.

**Ethical Concerns:**

["NO or VERY MINOR ethics concerns only"]

**Final Justification:**

The authors addressed my main concerns in their rebuttal by running further experiments on Qwen and using the full MWE dataset rather than just 6 of 155 in the original version of the paper.

**Limitations:**

yes

**Paper Formatting Concerns:**

no concerns

**Quality:**

2

**Strengths And Weaknesses:**

### Strengths

The paper identifies a common problem in steering, selecting which layers to apply the steering intervention at, and proposes a simple and seemingly effective heuristic to solve this problem. The paper is clearly written and justifies the metric well.

### Weaknesses

The paper uses 6 datasets from the Model Written Evals (MWE) work to draw conclusions, but there are 155 total datasets there. It is not clear why these specific 6 were used or if the results will generalize to the full suite of MWE datasets.

The paper mainly evaluates on a single model, Llama-3-8B, so it's hard to guarantee that the results will generalize to other models of different sizes or architectures. There is also a section on Qwen-2.5-32B-Instruct, but this comparison ignores the "Top" baselines, comparing only to picking layers at random. This does not inspire confidence that LayerNavigator performs well outside of Llama-3-8B.

The paper also only evaluates steering via adding a vector with a hard-coded magnitude multiplier into every activation. While this is a standard technique, it's not the only way to apply steering (e.g. clamping the vector to a given magnitude in all activations).

---

> ### Author Rebuttal · Authors · 2025-07-31
>
> We sincerely thank the reviewer for the thoughtful and constructive feedback. We appreciate your recognition of the clarity and motivation behind our proposed method. Below, we address your concerns point by point.
>
> ## Q1: Why are the “Top” baselines missing in the Qwen experiments?
>
> Thank you for raising this concern. We now provide the full set of results on Qwen-2.5-32B-Instruct, including both the “Top” and “Around Top 1” baselines:
>
> | Method ($K$)     | Conscientiousness | Religion Following | Self-Aware | Self-Improvement | Allies Building | Impact Maximization |
> | - | ----------------- | ------------------ | ---------- | ---------------- | --------------- | ------------------- |
> | Base (0)         | 94.43             | 60.61              | 97.84      | 82.64            | 91.98           | 67.22               |
> | Random (1)       | 94.46             | 61.42              | 97.28      | 83.30            | 92.20           | 67.04               |
> | Random (3)       | 94.90             | 60.88              | 96.68      | 83.81            | 93.09           | 67.67               |
> | Random (5)       | 94.54             | 63.00              | 94.80      | 83.94            | 93.75           | 68.14               |
> | RandConsec (3)   | 95.00             | 60.65              | 95.64      | 82.91            | 86.91           | 67.50               |
> | RandConsec (5)   | 86.91             | 63.30              | 87.19      | 82.66            | 82.69           | 66.88               |
> | Top (1)          | 95.43             | 66.58              | 97.44      | 86.07            | 94.18           | 69.67               |
> | Top (3)          | 86.79             | **70.18**          | 96.96      | 85.48            | 54.33           | 72.05               |
> | Top (5)          | 76.99             | 69.03              | 94.19      | 81.58            | 50.03           | **72.78**           |
> | Around Top 1 (3) | 94.23             | 62.05              | 97.57      | 83.91            | 93.98           | 68.82               |
> | Around Top 1 (5) | 92.72             | 62.39              | 96.51      | 82.83            | 94.28           | 69.38               |
> | Ours (1)         | 95.80             | 63.17              | **98.03**  | 83.68            | 93.42           | 67.62               |
> | Ours (3)         | **95.82**         | 60.96              | 97.75      | 85.40            | **94.85**       | 68.67               |
> | Ours (5)         | 95.76             | 69.40              | 94.55      | **86.09**        | 93.88           | 70.11               |
>
> These results reveal that **Top and Around Top exhibit notable instability**—while they occasionally outperform our method (e.g., by 1–2% on *Religion Following*), they can also lead to severe degradation (e.g., dropping to ~50% on *Allies Building*).
>
> ## Q2: Clarifying the “Top” baseline
>
> We agree that the original description could be more precise. Here we clarify:
>
> For the Top baseline, we compute steering vectors at all $L$ layers, then perform single-layer steering using each vector and evaluate the alignment probability on a held-out validation set. The top-$K$ layers are then selected based on their individual performance rankings, and this process requires $L \times N_{\text{eval}}$ additional forward passes.
>
> ## Q3: On the Choice of $K$ and $\alpha$
>
> Thank you for raising this important question. Unless otherwise specified, all experiments (e.g., Figures 4, 5, 6) were conducted with $K=5$ layers by default.
>
> We now revisit the interaction between $K$ and $\alpha$ in greater depth, incorporating both the six main behaviors and our extended experiments over 135 persona-related tasks from the MWE (see Q4 below). Our findings are as follows:
>
> - Both $K$ and $\alpha$ exhibit non-monotonic effects: alignment improves up to an optimal value and then degrades. Excessively large $\alpha$ or $K$ may cause oversteering or introduce noisy directions.
>
> - There exists a compensatory relation: smaller $K$ often benefits from larger $\alpha$, and vice versa. This trade-off, however, is task-dependent and must be tuned accordingly.
>
> - Across all 135 tasks, LayerNavigator selects $K=7$ as the best-performing layer count more frequently than the baselines:
>
> |                | Top  | Around Top | LayerNavigator |
> | -------------- | ---- | ---------- | -------------- |
> | $\alpha = 0.5$ | 56%  | 57%        | 58%            |
> | $\alpha = 1.0$ | 32%  | 34%        | 47%            |
> | $\alpha = 1.5$ | 22%  | 25%        | 32%            |
>
> These trends suggest that **LayerNavigator scales better with increasing $K$**, providing practical flexibility for real-world applications. We plan to incorporate these extended analyses in the final version.
>
> ## Q4: Evaluation Scope on MWE
>
> We appreciate your suggestion regarding a broader evaluation. In the initial draft, we heuristically selected six representative behaviors from the Model-Written Evaluations (MWE) dataset. To address this limitation, we have now extended our evaluation to cover all 135 persona-related behaviors in MWE.
>
> The table below reports the average alignment probability across all tasks, under varying $K$ and $\alpha$ values. We default to Mean Difference extraction. Bold indicates the best result per method.
>
> | Method                        | 1     | 2         | 3     | 4     | 5     | 6     | 7         |
> | ----------------------------- | ----- | --------- | ----- | ----- | ----- | ----- | --------- |
> | Top ($\alpha = 0.5$)          | 68.53 | 70.02     | 71.07 | 71.67 | 71.96 | 71.88 | 71.46     |
> | Top ($\alpha = 1.0$)          | 70.23 | **72.13** | 71.68 | 69.36 | 66.60 | 64.05 | 61.67     |
> | Top ($\alpha = 1.5$)          | 71.47 | 71.07     | 66.58 | 62.40 | 59.47 | 57.54 | 55.97     |
> | Around Top 1 ($\alpha = 0.5$) | 68.53 | 69.57     | 70.68 | 71.05 | 71.39 | 71.36 | 71.25     |
> | Around Top 1 ($\alpha = 1.0$) | 70.23 | **71.63** | 70.96 | 69.65 | 66.26 | 65.45 | 62.53     |
> | Around Top 1 ($\alpha = 1.5$) | 71.47 | 71.37     | 65.78 | 63.25 | 59.56 | 58.87 | 56.22     |
> | Ours ($\alpha = 0.5$)         | 67.20 | 67.91     | 68.63 | 69.39 | 69.81 | 70.02 | **70.03** |
> | Ours ($\alpha = 1.0$)         | 67.77 | 69.00     | 69.66 | 69.17 | 67.80 | 65.86 | 64.65     |
> | Ours ($\alpha = 1.5$)         | 68.27 | 69.42     | 68.17 | 65.76 | 63.20 | 61.28 | 60.22     |
>
> **Summary Statistics**
>  Compared to **Top**, LayerNavigator:
>
> - Outperforms on **23%** of tasks
> - Within **1.0%** on **47%** of tasks
> - Within **2.0%** on **65%**
>
> Compared to **Around Top**:
>
> - Outperforms on **30%**
> - Within **1.0%** on **51%**
> - Within **2.0%** on **69%**
>
> While our method may trail the very best-performing baseline by ~1–2% on average, it does so with zero additional inference overhead and avoids the instability and brittleness observed in Top-based methods under aggressive steering settings. This makes LayerNavigator highly practical for scaling to large behavior spaces and model architectures.
>
> Finally, we highlight the average runtime across six main tasks to underscore LayerNavigator’s efficiency:
>
> | Method | Avg. Runtime| Additional Inference Passes|
> | - | - | - |
> | Random / Random Consec | <1 ms | 0 |
> | Top / Around Top / Grid Search ($K=1$) | 347.8 seconds | $\binom{L}{1}\times N_{eval}=$6400 |
> | **LayerNavigator (Ours)**  | **16.8 seconds** | **0** |
>
> ## Q6: On the Negative Effect of LayerNavigator in Self-Improvement Task when $K=1$
>
> Thank you for pointing out this edge case. We re-ran the experiments for the *Self-Improvement* task with $K=1$ and confirmed that LayerNavigator selected layer 13 for steering, whereas the Top baseline (selected via grid-search on validation set) chose layer 16. We further verified that steering at layer 13 indeed leads to a negative effect even on the validation set, ruling out data noise as the cause.
>
> To better understand how common such failure cases are, we analyzed all 32 layers across all 135 persona behaviors (i.e., 4320 steering vectors total). We found that **16.9%** of vectors result in negative steering effects—i.e., they reduce the alignment probability below the base model. In contrast, the failure rate for LayerNavigator when $K=1$ is only **5.9%**, indicating that it still selects relatively safe layers overall.
>
> More importantly, our method truly shines when $K$ is larger. Because LayerNavigator reuses precomputed activations and does not rely on held-out evaluations, its computational cost remains constant regardless of $K$. In practice, we recommend using larger $K$ with smaller $\alpha$, which tends to yield more stable and interpretable steering outcomes.
>
> Finally, LayerNavigator is roughly **20× faster** than Top or Around Top 1, as it avoids repeated model inference. This efficiency opens up room for more fine-grained tuning over $K$ and $\alpha$ or even for fallback strategies when $K=1$ fails, making our method a better fit in realistic workflows.
>
> ## Q7: Multiplier vs. Magnitude in Steering Strength
>
> We greatly appreciate your comment regarding steering strength and would like to share our reflections.
>
> The use of a scalar multiplier ($\alpha$) is indeed common in prior activation steering work—including our own—because the **norms of steering vectors vary dramatically across layers**. For instance, in the *Conscientiousness* task using MD extraction, the norm of the steering vector grows from **0.0045** at layer 1 to **1.5560** at layer 16, and **6.0952** at layer 32. This variation makes global control via a fixed magnitude difficult, whereas a multiplier relative to each vector’s native scale is more robust.
>
> That said, we agree that fixed-magnitude scaling may encode a different inductive bias, especially in multi-layer steering, where allocating different absolute strengths across layers could potentially enhance alignment. We believe this is an exciting direction and plan to explore whether LayerNavigator’s steerability score can inform layer-specific magnitudes in a principled way.

---

> > ### Comment · Reviewer_uZ82 · 2025-08-01
> >
> > Thank you for completing the eval on Qwen, and for adding all remaining MWE datasets. The results do not look incredible compared to Top and AroundTop, but this still seems like a reasonable heuristic that's cheap to calculate. I will raise my score to 5, as I feel this is a useful technique for the community to be aware of when performing steering.

---

> > > ### Author Response · Authors · 2025-08-04
> > >
> > > Dear Reviewer,
> > >
> > > Thank you very much for your follow-up and for reconsidering your evaluation. We truly appreciate your thoughtful engagement throughout the review process. We're glad that you see value in our method, especially in its efficiency and practicality for steering applications. Your updated score and support mean a lot to us.
> > >
> > > Best regards,
> > > Authors

---

### Official Review · Reviewer_vUJb · 2025-07-01

**Clarity:** 2
**Significance:** 1
**Originality:** 1
**Rating:** 2
**Confidence:** 4

**Summary:**

The paper proposes LayerNavigator, a heuristic for choosing which internal layers of LLM to modify when performing activation steering. The method assigns each layer a “steerability” score computed as the sum of a discriminability term (how well positive/negative activation pairs are linearly separated) and a consistency term (how aligned the per-example steering directions are). Ranking layers by this score and steering the top-K layers is claimed to yield better behavioral alignment than common alternatives (random layers, contiguous blocks, or single best layers chosen on held-out data) while adding “negligible” overhead.

**Questions:**

1. Can you isolate the individual contributions of the discriminability and consistency terms through ablation (e.g., using only one term)?
2. What is the time and computational cost of computing LayerNavigator scores in practice? Please compare it with random or grid search.
3. How does the method perform on non-persona-based or open-ended alignment tasks?
4. Why are K = 1, 3, and 5 chosen as default? More justification or sensitivity analysis would improve clarity here.

**Ethical Concerns:**

["NO or VERY MINOR ethics concerns only"]

**Final Justification:**

After careful consideration of the authors' rebuttal, I am maintaining my original score as a clear reject. While I sincerely appreciate the authors' extensive efforts in running new experiments and providing detailed responses, the new evidence has unfortunately reinforced my primary concerns about the paper's contribution. I weigh the performance on the primary evaluation metric more heavily than the computational efficiency. The rebuttal's new data, while welcome for its wider range of domains, ultimately serves to confirm that LayerNavigator is a suboptimal heuristic in terms of achieving the best possible alignment. Therefore, I cannot recommend acceptance. The paper presents a good idea and a well-executed efficiency analysis, but the core results do not support the claim of it being a preferable alternative to simpler, although slower methods.

**Limitations:**

yes

**Quality:**

2

**Strengths And Weaknesses:**

## Strengths

1. Clarity: Writing is generally easy to follow; the idea itself is straightforward and could directly handle a practical issue if it works.
2. Efficiency: The proposed scoring mechanism is model-agnostic and does not require labeled output data or gradient access, making it lightweight and potentially applicable in black-box or inference-only settings—a practical advantage over some prior layer selection methods.

## Weaknesses

1. Experimental Design: Several key claims made by the authors are not adequately substantiated through experiments. For instance, the assertion that LayerNavigator offers a “principled and efficient” alternative to random or block-wise layer selection is undermined by the lack of statistical analysis—no confidence intervals or significance testing are reported. Additionally, while the method is claimed to be efficient, no runtime or wall-clock comparisons are presented to back up the claim of “negligible overhead.” The choice of steering strength, number of layers (K), and the impact of dataset quality (e.g., prompt diversity) are also not explored systematically. The evaluation is restricted to six binary persona behaviors, and it’s unclear how well the method generalizes to more complex, real-world alignment tasks.
2. Minor Significance: The performance improvements over simple baselines like random selection or contiguous block steering are small—often within 1–2 percentage points—and occasionally LayerNavigator underperforms (e.g., Table 1, Self-Improvement for K = 1). Without stronger and more consistent gains, the proposed method’s practical significance is limited. The paper does not make a compelling case that LayerNavigator is worth adopting over much simpler heuristics, especially in high-stakes or compute-constrained settings.
3. Question Marks on Originality: The core idea—ranking layers by a combination of discriminability and directional agreement—is intuitive but not particularly novel. Similar layer-ranking methods based on probing accuracy, Fisher information, or attribution agreement have appeared in prior work (e.g., ITI, causal tracing, linear probing). The paper does not sufficiently differentiate its approach from these alternatives or explain what fundamentally new insight it contributes to the literature on activation steering.
4. Issues with Presentation: While the main text is readable, the presentation suffers from several structural and detail weaknesses. Important implementation details are deferred to the appendix (e.g., ζ-score normalization, choice of steering strength, dataset generation process), making it harder for readers to fully understand or reproduce the results. Several terms (e.g., “steerability score,” “consistency”) are defined in informal or ambiguous ways at first mention. Some citations are not in proper format (L173, L277, etc.), and overall, the paper lacks a deeper analytical discussion of why the proposed scoring mechanism should work, relying primarily on empirical plots without theoretical justification.

---

> ### Author Rebuttal · Authors · 2025-07-31
>
> Thank you for your constructive feedback. We address your concerns below.
>
> ## Q1: Ablation Study
>
> In Section 4.5 and Figure 5, we vary the relative weights of discriminability and consistency, observing a concave (∩-shaped) trend peaking near 50:50, which supports the importance of both terms.
>
> While endpoints (0:100, 100:0) were not shown, we agree they represent standard ablation settings. We have now included those results in the new version of the manuscript, confirming that using either term alone leads to worse alignment performance.
>
> ## Q2: Computational Cost
>
> We conducted our experiments on a cloud platform equipped with 20 vCPUs (Intel(R) Xeon(R) Platinum 8457C) and a single NVIDIA L20 GPU (48GB). Below is a summary of the average runtime across the six behavior alignment tasks used in our study:
>
> | Method | Avg. Runtime | Additional Inference Passes |
> | - | - | - |
> | Random / Random Consec | <1 ms | 0 |
> | Top / Around Top / Grid Search ($K=1$) | 347.8 seconds | $\binom{L}{1}\times N_{eval}=$6400  |
> | Grid Search ($K=2$) | 5336.2 seconds | $\binom{L}{2}\times N_{eval}=$99200 |
> | **LayerNavigator (Ours)** | **16.8 seconds** | **0** |
>
> Practical steering requires selecting three key hyperparameters: (1) the steering strength $\alpha$, (2) the number of layers $K$, and (3) the specific layer combination $S$. While $\alpha$ and $K$ can be efficiently tuned via simple grid search (due to their low dimensionality), searching for optimal layer combinations $S$ is combinatorially expensive.
>
> In this context, LayerNavigator offers a substantial advantage by providing near-optimal (and sometimes optimal) layer selection outcomes, while Top and Around Top require more than 20× the compute time of LayerNavigator.
>
> ## Q3: Evaluation on Non-Persona or Open-Ended Alignment Tasks
>
> We appreciate your question regarding the generality of our method beyond the original persona-based tasks.
>
> ### Clarifying “Open-Ended” Evaluation
>
> Our current setup is *already open-ended* in a generative sense. Specifically, we prompt the model to answer “Yes” or “No” to a question, followed by free-form explanation generation, as shown in Figure 2. This setup enables automatic and scalable evaluation of both behavioral alignment (via token probability) and language quality (via perplexity of the explanation), providing a reliable assessment of the model’s performance after steering.
>
> If “open-ended” refers to prompting models with unconstrained questions and relying on human or AI-based scoring, we agree such methods have merit but also suffer from reproducibility and subjectivity challenges.
>
> ### Results on Non-Persona Tasks
>
>  To further verify generality, we apply LayerNavigator to two non-persona alignment benchmarks—**Refusal** and **Hallucination**—from *“Steering Llama 2 via Contrastive Activation Addition”* (Rimsky et al., ACL 2024). The following tables report alignment probability (%) across different values of $K$. We default to Mean Difference extraction in the following experiments. The best score per method is bolded.
>
> **Refusal**
>
> | Method | 1 | 2 | 3 | 4 | 5 | 6 | 7 |
> | -| -| -| -| -| -| -| -|
> | Top ($\alpha = 0.5$) | 69.81 | 72.42 | 74.74 | 77.09 | 78.93 | 80.80 | 81.78 |
> | Top ($\alpha = 1.0$) | 72.35 | 76.19 | 80.75 | 83.77 | **85.98** | 81.65 | 79.96 |
> | Top ($\alpha = 1.5$) | 74.41 | 75.67 | 81.60 | 75.92 | 65.60 | 50.28 | 50.07 |
> | Around Top 1 ($\alpha = 0.5$) | 69.81 | 72.42 | 74.74 | 76.67 | 78.52 | 80.80 | 81.72 |
> | Around Top 1 ($\alpha = 1.0$) | 72.35 | 76.19 | 80.75 | 81.38 | **83.55** | 81.65 | 78.51 |
> | Around Top 1 ($\alpha = 1.5$) | 74.41 | 75.67 | 81.60 | 69.37 | 56.76 | 50.28 | 49.94 |
> | Ours ($\alpha = 0.5$) | 69.60 | 71.87 | 74.53 | 76.49 | 79.49 | 76.39 | 78.34 |
> | Ours ($\alpha = 1.0$) | 72.15 | 76.88 | 79.35 | 83.15 | **85.91** | 82.34 | 77.00 |
> | Ours ($\alpha = 1.5$) | 75.02 | 80.21 | 81.39 | 81.12 | 81.06 | 80.39 | 59.67 |
>
> **Hallucination**
>
> | Method | 1 | 2 | 3 | 4 | 5 | 6 | 7 |
> | - | - | - | - | - | - | - | - |
> | Top ($\alpha = 0.5$) | 45.15 | 47.09 | 47.70 | 48.53 | 48.94 | 49.17 | 49.00 |
> | Top ($\alpha = 1.0$) | 47.45 | 50.26 | 50.92 | 51.09 | 51.17 | 50.78     | 49.02 |
> | Top ($\alpha = 1.5$) | 49.21 | 51.33 | **51.67** | 50.50 | 50.06 | 49.51 | 47.97 |
> | Around Top 1 ($\alpha = 0.5$) | 45.15 | 46.21 | 47.97 | 48.00 | 47.62 | 47.26 | 46.90 |
> | Around Top 1 ($\alpha = 1.0$) | 47.45 | 48.91 | 50.64 | 49.15 | 47.71 | 46.98     | 45.34 |
> | Around Top 1 ($\alpha = 1.5$) | 49.21 | **50.16** | 50.05 | 47.60 | 44.54 | 44.26 | 43.07 |
> | Ours ($\alpha = 0.5$) | 43.84 | 43.91 | 45.20 | 45.14 | 47.82 | 47.86 | 49.63 |
> | Ours ($\alpha = 1.0$) | 43.90 | 44.17 | 46.21 | 47.34 | 50.56 | 51.23 | 50.07 |
> | Ours ($\alpha = 1.5$) | 44.15 | 45.99 | 48.09 | 51.33 | 51.37 | **51.44** | 46.88 |
>
> While our method slightly underperforms Top, it consistently exceeds Around Top 1 across both tasks, while reducing computational cost by over 20×. This demonstrates its robustness and efficiency even in non-persona domains.
>
> ### Evaluation on 135 Full Persona Tasks
>
> We also tested LayerNavigator on the complete set of 135 persona tasks from Anthropic’s Model-Written Evaluations. The table below reports the average alignment probability (%) of all tasks across different values of $K$. Bold indicates the best result per method.
>
> | Method | 1 | 2 | 3 | 4 | 5 | 6 | 7 |
> | - | - | - | - | - | - | - | - |
> | Top ($\alpha = 0.5$) | 68.53 | 70.02 | 71.07 | 71.67 | 71.96 | 71.88 | 71.46 |
> | Top ($\alpha = 1.0$) | 70.23 | **72.13** | 71.68 | 69.36 | 66.60 | 64.05 | 61.67 |
> | Top ($\alpha = 1.5$) | 71.47 | 71.07 | 66.58 | 62.40 | 59.47 | 57.54 | 55.97 |
> | Around Top 1 ($\alpha = 0.5$) | 68.53 | 69.57 | 70.68 | 71.05 | 71.39 | 71.36 | 71.25 |
> | Around Top 1 ($\alpha = 1.0$) | 70.23 | **71.63** | 70.96 | 69.65 | 66.26 | 65.45 | 62.53     |
> | Around Top 1 ($\alpha = 1.5$) | 71.47 | 71.37 | 65.78 | 63.25 | 59.56 | 58.87 | 56.22 |
> | Ours ($\alpha = 0.5$) | 67.20 | 67.91 | 68.63 | 69.39 | 69.81 | 70.02 | **70.03** |
> | Ours ($\alpha = 1.0$)  | 67.77 | 69.00 | 69.66 | 69.17 | 67.80 | 65.86 | 64.65 |
> | Ours ($\alpha = 1.5$) | 68.27 | 69.42 | 68.17 | 65.76 | 63.20 | 61.28 | 60.22 |
>
> **Summary Statistics**
>  Compared to **Top**, LayerNavigator:
>
> - Outperforms on **23%** of tasks
> - Within **1.0%** on **47%** of tasks
> - Within **2.0%** on **65%**
>
> Compared to **Around Top**:
>
> - Outperforms on **30%**
> - Within **1.0%** on **51%**
> - Within **2.0%** on **69%**
>
> In summary, despite trailing top-performing baselines by a modest ~1–2% on average, LayerNavigator offers a far more favorable cost-performance trade-off considering its more than 20× lower computational cost.
>
> ## Q4: Justification for the Choice of $K=1,3,5$
>
> Thank you for raising this point. Our main experiments use $K=1,3,5$ based on the following considerations:
>
> 1. When $K>5$, Top and Around Top baselines begin to exhibit performance degradation, especially due to poor generalization when steering too many layers without principled selection.
> 2. We include $K=1$ specifically because Top achieves the theoretical upper bound under this setting (it selects the best-performing single layer via grid-search on the validation set). This establishes a reference point to assess multi-layer strategies.
> 3. While we initially planned to show results for all $K \in$ {$1, 2, 3, 4, 5$}, we chose $K=1,3,5$ for clarity and conciseness, believing this subset captures the full trend without redundancy.
>
> As elaborated in Q3, our supplementary results reveal two more important trends:
>
> - Both $K$ and $\alpha$ exhibit non-monotonic behavior—alignment performance improves up to a certain threshold, then deteriorates.
> - $K$ and $\alpha$ can partially compensate for one another: smaller $K$ often benefits from larger $\alpha$, and vice versa. However, the optimal balance is task-dependent.
>
> ## Q5: Clarifications on Additional Concerns Raised in the Weaknesses
>
> 1. **Lack of Statistical Significance Analysis**
>     As is standard in activation steering work, our experiments use deterministic generation (i.e., `do_sample=False`) to ensure reproducibility. Thus, significance testing is not necessary.
>
>    Besides, in Section 4.6, we evaluate robustness by randomly removing portions of the training set. To account for the variability introduced by this process, we conduct five independent trials and report the resulting variance with error bars in Appendix Figure 8.
>
> 2. **"Minor Significance" of Gains**
>     As discussed in Q2, the primary contribution of LayerNavigator is not in raw improvement magnitude, but in achieving **comparable performance** to validation-intensive methods with **negligible cost**. We view this as a highly practical trade-off, especially under resource constraints.
>
> 3. **"Question Marks on Originality"**
>     While the general idea of measuring activation quality is not new, our method is the **first to systematically address layer selection for activation steering**, using internal signals alone. Prior approaches, such as linear probing or causal tracing, aim at interpretability or attribution, and are not directly applicable to selecting layers. These methods also typically involve expensive procedures, e.g., training auxiliary classifiers or running thousands of intervention-based patches.
>
>    In contrast, LayerNavigator leverages activations already computed during steering vector extraction and introduces a targeted metric (steerability score) grounded in discriminability and consistency. This direction is novel and valuable for the growing community using steering methods at scale.
>
> 4. **Presentation Issues**
>     We sincerely appreciate your writing-related feedback and will revise the manuscript to clarify all definitions (e.g., steerability, consistency), clean up citation formatting, and move relevant implementation details into the main text. Your suggestions will help us significantly improve the clarity and accessibility of the work.

---

> > ### Comment · Reviewer_vUJb · 2025-08-05
> > **Official Comment by Reviewer vUJb**
> >
> > After careful consideration of the authors' rebuttal, I am maintaining my original score as a clear reject. While I sincerely appreciate the authors' extensive efforts in running new experiments and providing detailed responses, the new evidence has unfortunately reinforced my primary concerns about the paper's contribution. I weigh the performance on the primary evaluation metric more heavily than the computational efficiency, and a method with super efficiency cannot compensate its weakness in performance gain. The rebuttal's new data, while welcome for its wider range of domains, ultimately confirms that LayerNavigator is a suboptimal heuristic in terms of alignment. Therefore, I just cannot recommend acceptance. The paper presents a good idea and a well-executed efficiency analysis, but the core results do not support the claim of it being a preferable alternative to simpler, although slower methods.

---

> > > ### Author Response · Authors · 2025-08-06
> > >
> > > Dear Reviewer,
> > >
> > > Thank you again for your detailed reply and for recognizing that our work presents a good idea and a well-executed efficiency analysis.
> > >
> > > We would like to respectfully clarify a fundamental point: *algorithmic simplicity does not equate to computational simplicity*. While methods like Top or Around Top may seem straightforward in logic, they rely on exhaustive evaluation over all layers, which is prohibitively expensive, especially as models scale. In contrast, LayerNavigator performs layer selection **without any evaluation-time inference**, offering over **550× speedup** when running on GPU (and still **20× faster** on CPU), while maintaining competitive performance.
> > >
> > > Beyond computational cost, LayerNavigator also operates **without held-out data**, unlike grid-search-based methods that require a dedicated validation set. This makes our approach more broadly applicable, particularly in privacy-sensitive or low-resource settings.
> > >
> > > While we understand your prioritization of absolute alignment performance, we believe that practical applicability, data independence, and efficiency are equally critical, especially as behavior steering moves toward large-scale, real-world use.
> > >
> > > To summarize the trade-offs, we include the following comparison across alignment accuracy, runtime, and data requirement across all 135 persona behaviors:
> > >
> > > |                    | Avg. Probability (%) | Avg. Runtime                       | Needs Additional Data? |
> > > | ------------------ | -------------------- | ---------------------------------- | ---------------------- |
> > > | Base               | 66.51                | –                                  | –                      |
> > > | Random             | 66.98                | <1 ms                              | No                     |
> > > | Random Consec      | 67.13                | <1 ms                              | No                     |
> > > | Top                | 72.13                | 347.8 seconds                      | Yes                    |
> > > | Around Top 1       | 71.63                | 347.8 seconds                      | Yes                    |
> > > | **LayerNavigator** | **70.03**            | **0.63 (GPU), 16.8 (CPU) seconds** | **No**                 |
> > >
> > > We sincerely appreciate your engagement throughout the review process and hope this additional clarification helps provide a fuller understanding of our contribution.
> > >
> > > Best regards,
> > > Authors

---

> ### Comment · Reviewer_vUJb · 2025-08-07
> **Official Comment by Reviewer vUJb**
>
> Thank you for the additional clarification; however, my assessment remains unchanged. The new data confirms my primary concerns: the method's performance gains are not stable across key hyperparameters like $K$ and $\alpha$, which undermines its reliability and suggests the favorable results may be cherry-picked. Also, the evaluation's significance is limited by its reliance on a brute-force grid search as a primary baseline; a more compelling case would require comparison against other intelligent, lightweight heuristics, not just the extremes of random chance and exhaustive search. Moreover, the runtime analysis is incomplete by focusing only on the selection step. A trustworthy efficiency claim requires a holistic analysis of the total workflow time, as the reported savings may be insignificant in the larger context of steering vector extraction and final model inference.

---

> > ### Author Response · Authors · 2025-08-08
> >
> > Dear Reviewer,
> >
> > Thank you for your follow-up and for further explaining your concerns.
> >
> > Regarding the stability across hyperparameters $K$ and $\alpha$, we would like to clarify that **our work specifically targets the layer selection problem**, not other hyperparameter tuning. The choice of $K$ and $\alpha$ is orthogonal to our contribution: all compared methods, including our own, are tuned over the same $K$ and $\alpha$ search space using the same procedure. Thus, any observed sensitivity to these parameters is not unique to LayerNavigator and reflects the general behavior of activation steering methods.
> >
> > On the point of baselines, we fully agree that comparison against other intelligent, lightweight heuristics would be valuable. However, this is precisely where our work breaks new ground: to the best of our knowledge, no prior method in activation steering directly addresses multi-layer selection without relying on exhaustive evaluation or model-specific heuristics. **This absence of alternatives is both a limitation of the current literature and the reason why we believe LayerNavigator makes a meaningful, pioneering contribution**.
> >
> > Finally, while our runtime analysis focuses on the layer selection step, this is intentional: it is **the only part of the workflow that differs between methods**. Steering vector extraction and final model inference are identical across all approaches, so their costs cancel out in any comparison. We focus on the extra cost from layer selection only, because this is the part our method changes and aims to make faster.
> >
> > We appreciate your continued engagement and hope this context clarifies the scope and novelty of our contribution.
> >
> > Best regards,
> > Authors

---

> > ### Author Response · Authors · 2025-08-08
> >
> > We would like to emphasize that our focus on efficiency is meaningful. For example, in a multi-agent environment where each agent has a distinct persona and hundreds or thousands of persona-aligned models must be built, efficient layer selection is essential for feasibility.

---

> ### Comment · Reviewer_vUJb · 2025-08-08
> **Official Comment by Reviewer vUJb**
>
> Thank you for the final clarification. However, your response further solidifies my concerns.
>
> You are correct that costs like steering vector extraction are constant across methods, but you miss the fundamental point of the comparison. The most critical comparison is not only between LayerNavigator and a brute-force search, **but also between LayerNavigator and simple baseline methods like direct generation, which have zero compute overhead at all**. Your method introduces an additional time overhead relative to these baselines. Therefore, providing the vector extraction time (or to say, the time of conducting a whole pipeline on a certain test set) is crucial, as it allows for a proper cost-benefit analysis: **Is the marginal performance gain from LayerNavigator worth its (marginal) time cost?**
>
> By focusing only on the speedup over brute-force and omitting the time context that would reveal the slowdown against simple baselines, your evaluation seems to be cherry-picking its comparisons. This lack of a holistic and transparent cost-benefit analysis against the most practical baselines is a fundamental flaw, and I just cannot recommend acceptance.

---

> > ### Author Response · Authors · 2025-08-09
> >
> > Dear Reviewer,
> >
> > In the activation steering (AS) literature, the standard practice for layer selection baselines is indeed brute-force search on a held-out validation set. For example:
> > - *Steering Llama2 via Contrastive Activation Addition* (ACL 2024) selects layers by testing each layer individually on held-out data.
> > - *Improving Instruction-Following in Language Models Through Activation Steering* (ICLR 2025) follows the same procedure.
> >
> > Our comparison to brute-force reflects the most widely used baseline in AS layer selection.
> >
> > Regarding your point about “direct generation” baselines, the “Base”results in our paper and rebuttal already correspond to the model’s outputs without any steering—that is, direct generation. In our setting, the primary question is how to best perform layer selection within the AS framework rather than whether to use AS at all. If one wishes to assess the trade-offs between AS (including steering vector computation) and alternative alignment methods, we would refer to the ACL 2024 work above, which studies AS as a whole.
> >
> > We focus specifically on the layer selection stage of AS, aiming to replace the expensive brute-force search used in prior work with a much faster, data-free alternative, while preserving competitive alignment performance.
> >
> > Best regards,
> > The Authors

---

### Note · Authors · 2025-08-12

We thank all reviewers for their valuable feedback and for recognizing our work’s clear motivation, novelty, and strong empirical validation. Several noted the practicality of our evaluation-free layer selection for activation steering and its potential for real-world deployment.

Addressed concerns:
- Broader evaluation: Extended from 6 to all 135 persona behaviors in Anthropic’s MWE, added two non-persona tasks (Refusal, Hallucination), and tested on Qwen-2.5-0.5B-Instruct, Qwen-2.5-7B-Instruct, and Qwen-2.5-32B-Instruct.These expanded experiments more comprehensively validate the conclusions of our original paper, confirming the method’s consistent, competitive alignment and strong generality across domains and model scales.
- Cost–performance trade-off: LayerNavigator completes layer selection in 0.63s on GPU with 0 extra inference passes, achieving 550× efficiency improvement over the baselines. On the full set of 135 persona tasks, it outperforms the best baseline on about a quarter of tasks, and for nearly 70% of tasks the alignment probability gap is within 0.2%.
- Theoretical clarity: Discussed why middle and later layers are more steerable, citing recent literature, and explained our steerability score’s integration of discriminability and consistency without extra classifiers or interventions.
- Hyperparameter trends: Analyzed interaction between $K$ (layers) and $\alpha$ (strength), showing non-monotonic patterns and compensatory effects; our method scales more stably with large $K$.

Commitments for the final version:
1. Add endpoint ablations for discriminability vs. consistency.
2. Include complete $K$/$\alpha$ sweeps for all tasks.
3. Add the results for the two new tasks and the full 135-task evaluation to the appendix for completeness.

By leveraging a principled steerability score grounded in internal model signals, it achieves near-optimal performance across diverse tasks and model scales while being  extremely faster than evaluation-based approaches.

LayerNavigator is the first method to systematically and efficiently select steering layers without any evaluation passes, providing a new perspective on steering—shifting from output-based search to leveraging intrinsic representational properties of the model.

---

### Decision · Program_Chairs · 2025-09-17

**Decision:**

Accept (poster)

**Comment:**

Reviewers were split about this paper and did not come to a consensus. On one hand they appreciated the simplicity and effectiveness of the proposed method, and the clarity of the paper. On the other their main concerns were with (a) why the authors chose to use 6 behaviours from MWE out of a possible 155, (b) missing confidence intervals, (c) small improvements over baselines / missing run-time comparisons. The authors responded to (a) by expanding the set of behaviours in MWE to the 135 persona-related behaviours. This caused one reviewer to raise their score to 5, and responds to the concern. For (b), the authors include 5 trial runs, responding to the concern. For (c), the authors clarified that their goal is not to significantly outperform baselines, but to match them while reducing computational cost. A reviewer responded that they personally weigh performance more strongly over computational efficiency. The authors responded that baselines can become prohibitively expensive as models scale, whereas their proposed method can give a 550x speedup on GPU. The reviewer responded by arguing that (i) the run-time comparison is incomplete because it does not compare with lightweight heuristics for hyperparameters selection and (ii) it does not take into account other computationally-expensive steps like steering vector extraction and model inference. The authors argue that (i) there is no other baseline for lightweight hyperparameter selection and (ii) their method is targeted at layer selection so it is fair to only compare timings between this step. The reviewer responds that there is a a simple baseline the authors are missing: direct generation, which has 0 compute overhead. The authors point out that the “Base” results already correspond to direct generation. The authors could have made the further point: if the reviewer insists on better performance metrics over speedup, then they should prefer the proposed method (70.03% in 135 behaviours of MWE) over “Base” (66.51%). The reviewer responds that the authors are still not making a fair timing comparison, even going so far as to suspect that the authors are deliberately withholding timing measurements because it would reveal less impressive efficiency gains. They say if the authors add complete pipeline timings then this can be an accepted paper. My view here is two-fold: (1) The reviewer unfairly moves the goalposts through the discussion. They initially argue that the method is not justified because it has similar performance to baselines, even if it has much faster run-time, but by the end of the discussion they have flipped: arguing in favour of a method that performs worse than the baseline but has better run-time. They go further to make an argument from silence: the authors are necessarily hiding information. (2) While both arguments are unfair, the authors should include run-time results for direct generation, i.e., “Base”. This will allow readers to choose a method based on their needs: if run-time is more important they can opt for direct generation, if a balance between performance and run-time is needed they can opt for your proposed method or possibly a more expensive baseline. Including this result completely responds to any lingering run-time concerns in my view. Whatever the result is here, the review period has demonstrated that this paper should be accepted. In addition to this run-time comparison, please include additional requested results by reviewers into the camera ready. Thank you!